# NEIL1 and NEIL2 DNA glycosylases protect neural crest development against mitochondrial oxidative stress

Dandan Han[1†], Lars Schomacher[1†*], Katrin M Schüle[1†], Medhavi Mallick[1], Michael U Musheev[1], Emil Karaulanov[1], Laura Krebs[1], Annika von Seggern[1], Christof Niehrs[1,2*]

[1]Institute of Molecular Biology (IMB), Mainz, Germany; [2]Division of Molecular Embryology, DKFZ-ZMBH Alliance, Heidelberg, Germany

**Abstract** Base excision repair (BER) functions not only in the maintenance of genomic integrity but also in active DNA demethylation and epigenetic gene regulation. This dual role raises the question if phenotypic abnormalities resulting from deficiency of BER factors are due to DNA damage or impaired DNA demethylation. Here we investigate the bifunctional DNA glycosylases/ lyases NEIL1 and NEIL2, which act in repair of oxidative lesions and in epigenetic demethylation. *Neil*-deficiency in *Xenopus* embryos and differentiating mouse embryonic stem cells (mESCs) leads to a surprisingly restricted defect in cranial neural crest cell (cNCC) development. *Neil*-deficiency elicits an oxidative stress-induced TP53-dependent DNA damage response, which impairs early cNCC specification. Epistasis experiments with *Tdg*-deficient mESCs show no involvement of epigenetic DNA demethylation. Instead, *Neil*-deficiency results in oxidative damage specific to mitochondrial DNA, which triggers a TP53-mediated intrinsic apoptosis. Thus, NEIL1 and NEIL2 DNA glycosylases protect mitochondrial DNA against oxidative damage during neural crest differentiation.

DOI: https://doi.org/10.7554/eLife.49044.001

*For correspondence:
l.schomacher@imb-mainz.de (LS);
C.Niehrs@imb-mainz.de (CN)

†These authors contributed equally to this work

Competing interests: The authors declare that no competing interests exist.

## Introduction

DNA repair is crucial to maintain genomic integrity in the face of exogenous and endogenous challenges. Cells express an arsenal of DNA repair enzymes that maintain genomic integrity, and mouse mutants have revealed a critical role of DNA repair in both organismic ageing and disease (*Jacobs and Schär, 2012*; *Lombard et al., 2005*). In addition, there is now compelling evidence that DNA repair enzymes function not only in lesion control but have been co-opted in epigenetic gene regulation via active DNA demethylation (*Bellacosa and Drohat, 2015*; *Schuermann et al., 2016*; *Wu and Zhang, 2017*). The best understood active demethylation mechanism involves TET dioxygenases, which iteratively oxidize the methyl group at C5 to yield 5-hydroxymethylcytosine (5hmC) (*Kriaucionis and Heintz, 2009*; *Tahiliani et al., 2009*), 5-formylcytosine (5fC) (*Ito et al., 2011*; *Pfaffeneder et al., 2011*), and 5-carboxylcytosine (5caC) (*He et al., 2011*; *Ito et al., 2011*). Thymine DNA glycosylase (TDG) excises 5fC and 5caC and the ensuing abasic site intermediate is processed by BER to restore unmethylated C (*Cortázar et al., 2011*; *Cortellino et al., 2011*; *He et al., 2011*; *Maiti and Drohat, 2011*; *Shen et al., 2013*; *Song et al., 2013*). The need for abasic site processing during active DNA demethylation therefore places BER enzymes center stage in epigenetic gene regulation.

Deficiency of the BER enzymes (e.g. TDG, APEX1, POLB, LIG3, XRCC1) can lead to abnormalities or lethality during embryogenesis (*Cortázar et al., 2011*; *Cortellino et al., 2011*; *Puebla-Osorio et al., 2006*; *Sugo et al., 2000*; *Tebbs et al., 1999*; *Xanthoudakis et al., 1996*), but the

**eLife digest** The face of animals with a backbone is formed in great part by a group of cells called cranial neural crest cells. When too few of these cells are made, the skull and the face can become deformed. For example, the jaw- or cheekbones can be underdeveloped or there may be defects in the eyes or ears. These types of abnormalities are among the most common birth defects known in humans.

NEIL1 and NEIL2 are mouse proteins with two roles. On the one hand, they help protect DNA from damage by acting as so-called 'base excision repair enzymes', meaning they remove damaged building blocks of DNA. On the other hand, they help remove a chemical group known as a methyl from DNA building blocks in a process called demethylation, which is involved both in development and disease. Previous research by Schomacher et al. in 2016 showed that, in frogs, the absence of a similar protein called Neil2, leads to deformities of the face and skull.

Han et al. – who include some of the researchers involved in the 2016 study – have now used frog embryos and mouse embryonic stem cells to examine the role of the NEIL proteins in cranial neural crest cells. Stem cells can become any type of cell in the body, but when NEIL1 and NEIL2 are missing, these cells lose the ability to become cranial neural crest cells.

To determine whether the effects of removing NEIL1 and NEIL2 were due to their role in DNA damage repair or demethylation, Han et al. removed two proteins, each involved in one of the two processes. Removing APEX1, which is involved in DNA damage repair, had similar effects to the removal of NEIL1 and NEIL2, while removing TDG, which only works in demethylation, did not. This indicates that NEIL1 and NEIL2's role in DNA damage repair is likely necessary for stem cells to become cranial neural crest cells.

Although NEIL1 and NEIL2 are part of the DNA repair machinery, Han et al. showed that when stem cells turn into cranial neural crest cells, these proteins are not protecting the cell's genomic DNA. Instead, they are active in the mitochondria, the compartments of the cell responsible for producing energy, which have their own DNA. Mitochondria use oxygen to produce energy, but by-products of these reactions damage mitochondrial DNA, explaining why mitochondria need NEIL1 and NEIL2. These results suggest that antioxidants, which are molecules that protect the cells from the damaging oxygen derivatives, may help prevent deformities in the face and skull. This theory could be tested using mice that do not produce proteins involved in base excision repair, which could be derived from the cells lacking NEIL1 and NEIL2.

DOI: https://doi.org/10.7554/eLife.49044.002

etiology of the physiological defects is often poorly understood. Notably, it is unclear if the phenotypic abnormalities are due to accumulating DNA damage or impaired DNA demethylation.

An example for BER enzymes acting both in lesion control and in epigenetic gene regulation are the endonuclease VIII-like glycosylases 1 and 2 (NEIL1 and NEIL2). These enzymes process oxidative DNA base lesions (*Bandaru et al., 2002*; *Hazra et al., 2002a*; *Hazra et al., 2002b*; *Takao et al., 2002*), but recently they have also been implicated in the machinery that removes 5-methylcytosine (5mC) from DNA during epigenetic DNA demethylation (*Müller et al., 2014*; *Schomacher et al., 2016*; *Slyvka et al., 2017*; *Spruijt et al., 2013*). NEIL1 and NEIL2 are bifunctional enzymes, which not only excise the damaged base but introduce a DNA single strand break via their AP lyase activity (*Hazra et al., 2002a*; *Hazra et al., 2002b*), while NEIL3 is mainly a monofunctional DNA glycosylase (*Krokeide et al., 2013*). NEIL1 is involved in prereplicative repair during S-phase (*Hegde et al., 2013*), and NEIL2 preferentially processes oxidized bases from transcribing genes via transcription-coupled BER (*Banerjee et al., 2011*). During epigenetic DNA demethylation NEIL1 and NEIL2 cooperate with TDG to excise oxidized 5mC intermediates generated by TET enzymes (*Schomacher et al., 2016*; *Slyvka et al., 2017*).

Mice deficient of NEIL1 are viable but display metabolic syndrome and brain dysfunction (*Canugovi et al., 2012*; *Rolseth et al., 2017*; *Vartanian et al., 2006*). *Neil2* null mice are also viable but are susceptible to inflammation (*Chakraborty et al., 2015*), while Neil2-deficient frog embryos display neural crest defects (*Schomacher et al., 2016*). This raises the question why and how does a defect in NEIL DNA glycosylases lead to these diverse and tissue-specific phenotypes? Both

epigenetic regulation and DNA damage can, in principle, impact neural crest development (*Hu et al., 2014*; *Sakai and Trainor, 2016*; *Simões-Costa and Bronner, 2015*), thus what is the relative contribution of oxidative lesion control and epigenetic DNA demethylation to the NEIL phenotypes?

To address these questions we have investigated Neil-deficient *Xenopus* embryos and created and characterized seven mouse embryonic stem cell (mESC) lines deficient for *Neil1,2,3* (triple and single knockouts), *AP-endonuclease 1* (*Apex1*), *Thymine DNA glycosylase* (*Tdg*) and *Neil1/Tdg*. We describe a mechanism where NEIL-deficiency elicits an oxidative stress-induced, TP53-dependent DNA damage response (DDR), which induces apoptosis and impairs early cNCC specification. We show that *Neil1-* and *Neil2*-deficiency leads to accumulation of oxidative DNA damage in mitochondria. Our work demonstrates how impaired removal of oxidative lesions can lead to a selective lineage defect during embryonic development. Our study contributes to the understanding of aberrant cNCC development, the root cause of congenital craniofacial malformations (*Wilkie and Morriss-Kay, 2001*).

## Results

### Neil2-deficiency induces a Tp53 DNA damage response in *Xenopus* embryos

We showed previously that in *Xenopus* embryos knockdown of Neil2 with an antisense morpholino oligonucleotide (*neil2* MO) induces head and tail abnormalities at tailbud stage, which are caused by impaired cranial neural crest cell (cNCC) specification at neurula stage (*Schomacher et al., 2016*). Of note, MOs are the loss-of-function approach of choice in model systems with large maternal stores of mRNA such as *Xenopus*, which can be targeted by MOs but not by for example TALEN or CRISPR/Cas9 approaches (*Blum et al., 2015*; *El-Brolosy et al., 2019*; *El-Brolosy and Stainier, 2017*; *Rossi et al., 2015*). The specificity of the *neil2* MO had been documented (*Schomacher et al., 2016*) i) by phenocopy with a second non-overlapping morpholino, and ii) by rescue of the head and tail abnormalities with orthologous human *NEIL2* mRNA, which was not targeted by *neil2* MO (*Figure 1A*).

Proper differentiation of cNCCs is crucial for the development of craniofacial cartilage and bone structures (*Sakai and Trainor, 2016*). Indeed, Alcian blue staining of head cartilage in *neil2* MO-injected embryos (single blastomere injection at 2-cell stage) revealed head cartilage, notably branchial cartilage defects at tadpole stage (*Figure 1B*), which were rescued by simultaneous injection of human *NEIL2* mRNA (*Figure 1B*).

To gain insight into the underlying mechanism leading to the cNCC phenotype, we microinjected *Xenopus* embryos with *neil2* MO and carried out RNA-seq gene expression profiling at early tailbud stage (*Supplementary file 1*). Differential expression analysis yielded a similar number of a few hundred up- and downregulated genes (*Figure 1C*). Interestingly, pathway enrichment analysis revealed significant results only for the upregulated genes with the top hits 'Tp53 pathway' and 'Tp53 pathway feedback loops' suggestive of a DNA damage response (DDR) (*Figure 1D*). Indeed, protein levels of both Tp53 and its upstream regulator phospho-Chk1 (pChk1) were induced in neural plates of Neil2 morphants, indicative for a DDR (*Figure 1E*). RT-qPCR confirmed upregulation of *tp53* and its target genes, including *ccng1*, *eda2r*, *aen*, and *riok3* (*Figure 1F*). Furthermore, co-injection of human *NEIL2* mRNA rescued induced pChk1 in Neil2 morphants, ruling out an unspecific Tp53-response to MO injection (*Figure 1G*). Notably, human *NEIL2* mRNA also reduced basal pChk1 levels.

The DDR in Neil2 morphants induced direct Tp53 targets characteristic for apoptosis (e.g. *eda2r*, *aen*). Apoptosis is linked to cNCC developmental defects since ablation of Tp53 and concomitant block of apoptosis suppress cranial facial abnormalities in *Tcof1* mouse mutants (*Jones et al., 2008*). Indeed, while cell proliferation seemed unaffected in *neil2* MO-injected embryos as judged by phospho-histone H3 levels (*Figure 1H*), we observed elevated Caspase-3 cleavage in dissected neural plates but less in non-neural tissue (*Figure 1I*), indicative of cNCC-specific apoptosis in Neil2 morphants. Consistently, human *NEIL2* mRNA injection in Neil2 morphants rescued elevated Caspase-3 cleavage to endogenous levels, corroborating the specificity of the *neil2* MO-induced apoptosis in neural plates. Human *NEIL2* mRNA injection also decreased basal levels of cleaved Caspase-3, both in neural plates and non-neural tissue (*Figure 1I*), suggesting that elevated *Neil2* expression protects

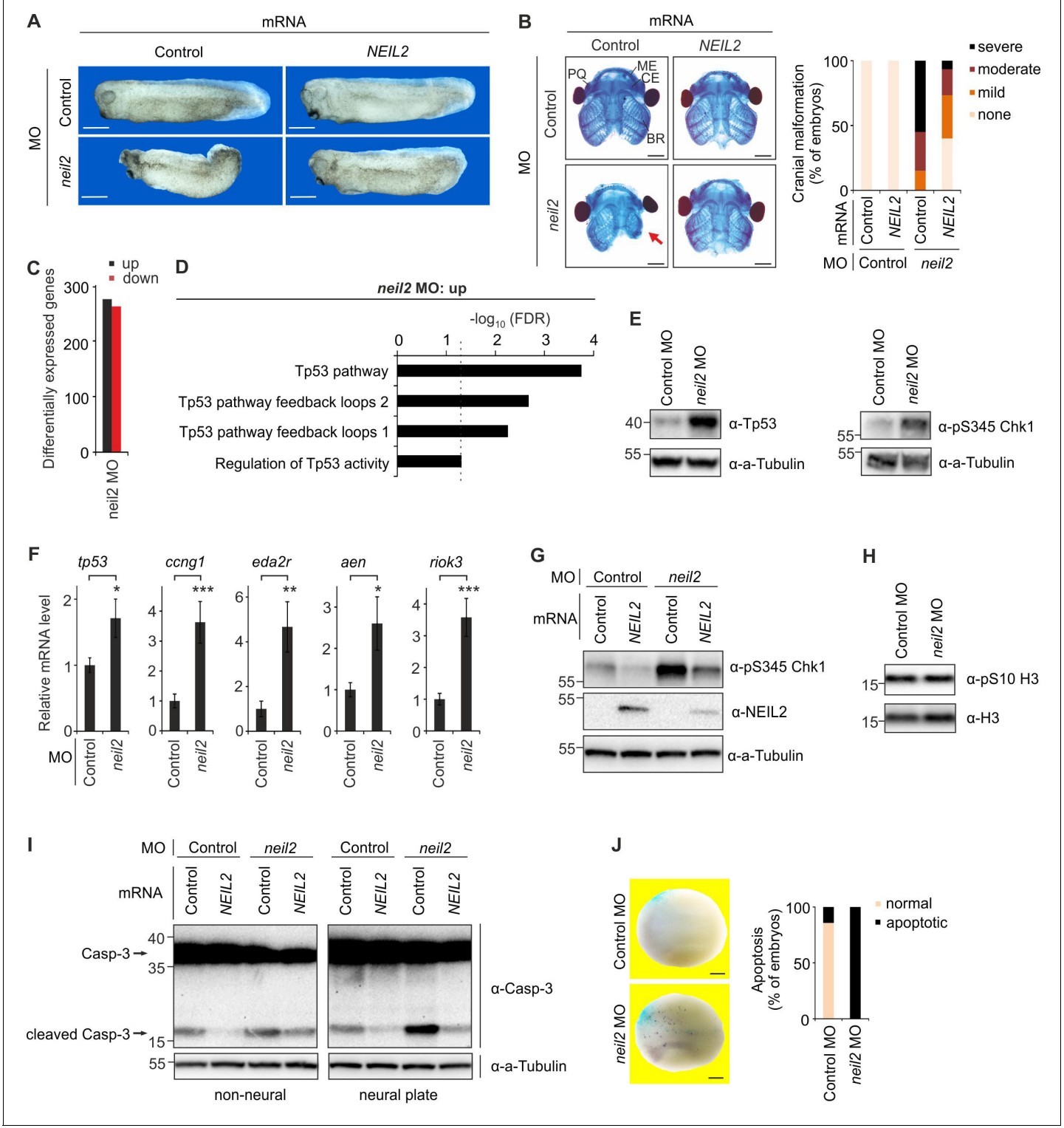

**Figure 1.** Neil2 protects against a Tp53 DNA damage response-induced apoptosis in *Xenopus* neuroectoderm. (**A**) Phenotypes at stage 32 of *Xenopus leavis* embryos injected at one-cell stage with control- or *neil2* MO (left). Human *NEIL2* or bovine *Preprolactin* (control) mRNA was co-injected for rescue experiments (right). Scale bars, 500 µm. (**B**) Left, representative stainings of cranial cartilage of stage 45 embryos unilaterally injected with control or *neil2* MO, and control or human *NEIL2* mRNA for rescue purpose as indicated. PQ, palatoquadrate cartilage; ME, Meckel's cartilage; CE, ceratohyal cartilage; BR, branchial cartilage. Arrow indicates cartilage defects in Neil2 morphants. Scale bars, 500 µM. Right, quantification of embryo malformations (n = 15, 21, 20 and 15 embryos per group, from left to right). (**C**) Quantification of differentially expressed genes at stage 23 of *neil2* MO-

*Figure 1 continued on next page*

*Figure 1 continued*

injected embryos. (D) Pathway enrichment analysis of *neil2* MO upregulated genes. Dashed line indicates the significance threshold FDR = 0.05. (E) Western blot for total Tp53 and phosphoserine (pS345) Chk1 in dissected stage 14 neural plates of control and *neil2* MO-injected embryos. Alpha (a-) Tubulin served as loading control. Molecular weight of marker proteins [x10$^{-3}$] is indicated on the left. Note that *X. laevis* Tp53 migrates at ~43 kDa. (F) qPCR expression analysis of *tp53* and Tp53 target genes in control- and *neil2* MO-injected embryos at stage 14. Expression of examined genes was normalized to *h4* and is presented relative to mRNA levels in control MO-injected embryos (mean ±s.d., n = 3 embryo batches consisting of 6 embryos each). (G) Western blot for pS345 Chk1 and NEIL2 in neural plates from control and *neil2* MO-injected stage 14 embryos co-injected with control or human *NEIL2* mRNA. Alpha (a-) Tubulin served as loading control. (H) Western blot for phosphoserine (pS10) histone H3 in neural plates from control and *neil2* MO-injected embryos at stage 15. Total histone H3 served as loading control. (I) Western blot analysis for Caspase-3 in dissected non-neural tissue and neural plates from control and *neil2* MO-injected stage 14 embryos co-injected with control and human *NEIL2* mRNA. Alpha (a-) Tubulin served as loading control. Uncleaved and cleaved (active) Caspase-3 are indicated (arrows). (J) TUNEL (apoptosis) assay of stage 16 embryos injected unilaterally with *neil2* MO and *lacZ* lineage tracer (TUNEL, dark blue speckles; *lacZ*, light blue speckles). Scale bars, 200 µM. Right, quantification of TUNEL signal (n = 7 and 10 embryos per group, from left to right).
DOI: https://doi.org/10.7554/eLife.49044.003

against apoptosis in whole embryos. Whole mount TUNEL assay of unilaterally *neil2* MO-injected embryos confirmed elevated apoptosis by *Neil2*-deficiency in stage 16 embryos (*Figure 1J*). We conclude that Neil2 protects against Tp53-mediated cell apoptosis in *Xenopus* embryos, notably in neural plate tissue.

## Intrinsic apoptosis triggers malformations in Neil2-deficient *Xenopus* embryos

To test if the phenotypic malformations in Neil2-morphants are related to elevated apoptosis, we blocked the apoptosis pathway by co-injection with *bcl2l1* mRNA. *Bcl2l1* is a *Xenopus* homologue of mammalian BCL2, an anti-apoptotic factor acting downstream of TP53 (*Hemann and Lowe, 2006*; *Tsujimoto, 1998*). Importantly, *bcl2l1* overexpression substantially reduced phenotypic abnormalities of Neil2 morphants (*Figure 2A*), and rescued elevated cleaved Caspase-3 levels in a dose-dependent manner (*Figure 2B*). As expected, *bcl2l1* expression had no effect on endogenous nor induced Tp53 protein levels (*Figure 2B*).

BCL2 regulates the intrinsic apoptosis pathway that is associated with mitochondria and results in mitochondrial dysfunction. Upregulation of mitochondrial (mt) gene expression is a characteristic response to mitochondrial dysfunction (*Heddi et al., 1999*; *Reinecke et al., 2009*). Indeed, we observed increased expression of *mt-Nd1,4* and *5* (*NADH dehydrogenase 1,4* and *5*) and *mt-co3* (*cytochrome c oxidase III*) in Neil2-morphant whole embryos, supporting ongoing intrinsic apoptosis and mitochondrial dysfunction (*Figure 2C*).

## Oxidative stress causes neural crest defects in Neil2-deficient *Xenopus* embryos

What leads to the induction of a Tp53 DDR in Neil2 morphants? NEIL2 processes oxidative base lesions induced by reactive oxygen species (ROS), such as 8-oxoguanine (8oxoG), 5-hydroxyuracil (5hU), thymine glycol, and the formamidopyrimidines FapyG and FapyA (*Hailer et al., 2005*; *Jacobs and Schär, 2012*), suggesting that accumulation of ROS DNA damage may account for embryonic abnormalities in Neil2-deficient *Xenopus* embryos. We therefore analyzed if *Xenopus* embryos are competent to mount a DDR following oxidative damage and if a DDR results in developmental abnormalities. Embryo treatment with the ROS producer pyocyanin (*Zhao et al., 2014*) upregulated Tp53 and pChk1 protein (*Figure 3A*), induced expression of *tp53* and its target genes (*Figure 3B*), and elevated Caspase-3 cleavage (*Figure 3C*). Moreover, pyocyanin-treated embryos phenocopied Neil2 morphants, displaying similar head and tail abnormalities (*Figure 3D*). Neurula stage embryos showed cNCC specification defects, where the cNCC markers *sox10*, *twist*, and *snail2* were downregulated, while the markers *sox3* (pan-neural), *en2* (midbrain), and *rx1* (eye) were unaffected (*Figure 3E*). Elevated ROS levels sensitized Neil2 morphants since pyocyanin treatment of embryos injected with a sub-critical dose of *neil2* MO, which alone did not yield abnormalities, elicited exacerbated abnormalities (*Figure 3F*). We conclude that *Neil2*-deficiency and ROS damage induce a Tp53 DDR in *Xenopus* embryos, leading to cNCC defects.

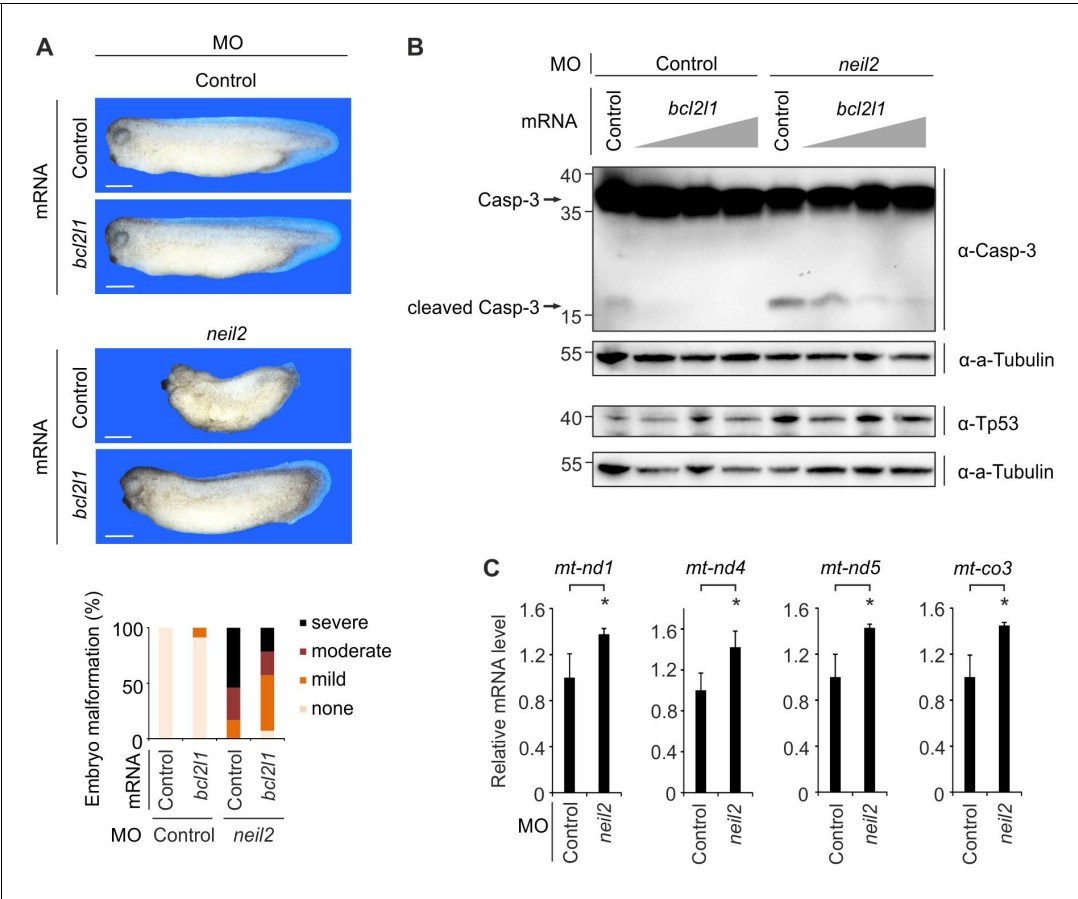

**Figure 2.** Malformations in Neil2-deficient *Xenopus* embryos are mediated by intrinsic apoptosis. (A) Top, representative phenotypes at stage 32 of *Xenopus laevis* embryos injected at one-cell stage with control- or *neil2* MO (bottom). *Xenopus bcl2l1* or bovine *Preprolactin* (control) mRNA was co-injected for rescue purpose. Scale bars, 500 μm. Bottom, quantification of embryo malformations (n = 17, 23, 24 and 14 embryos per group, from left to right). (B) Western blot for Caspase-3 (top) and Tp53 (bottom) in neural plates from control and *neil2* MO-injected stage 14 embryos co-injected with control or increasing amounts of *bcl2l1* mRNA (0.5, 1 and 2 ng). Alpha (a-) Tubulin served as loading control. Uncleaved and cleaved (active) Caspase-3 are indicated (arrows). Molecular weight of marker proteins [x10$^{-3}$] is indicated on the left. (C) qPCR expression analysis of mitochondrial (mt) genes in control- and *neil2* MO-injected embryos at stage 14. Expression of mt-genes was normalized to *h4* and is presented relative to mRNA levels in control MO-injected embryos (mean ±s.d., n = 3 embryo batches consisting of 6 embryos each).
DOI: https://doi.org/10.7554/eLife.49044.004

Importantly, treatment of embryos with Vitamin C, a prominent antioxidant (*Arrigoni and De Tullio, 2002*), attenuated the severe malformations of Neil2 morphants (*Figure 3G*), supporting ROS as the basis of cNCC defects in the absence of Neil2.

## A neuroectoderm-restricted Tp53 response triggers neural crest defects in *Xenopus* embryos

The Tp53 response to DNA damage is a widespread cellular phenomenon (*Ciccia and Elledge, 2010*). Hence, what restricts the DDR mostly to neuroectoderm during *Xenopus* development? Among the most upregulated genes in Neil2 morphants was the direct Tp53 target gene *cyclin-G1* (*ccng1*) (*Supplementary file 1*) (*Okamoto and Beach, 1994*). Ccng1 interacts with Mdm2, another Tp53 target gene, and is involved in a negative feedback loop regulating Tp53 protein levels induced by DNA damage (*Kimura and Nojima, 2002*; *Okamoto et al., 2002*). Unilateral injection of *neil2* MO in *Xenopus* embryos with the lineage tracer β-galactosidase confirmed upregulation of *ccng1* on the injected side (*Figure 4A*). *Ccng1* induction was restricted to the neural plate even in embryos where the lineage tracer extended to mesoderm and endoderm (*Figure 4A*). To test if the spatial restriction of *ccng1* expression reflects tissue-specificity of the DDR, we provoked a systemic

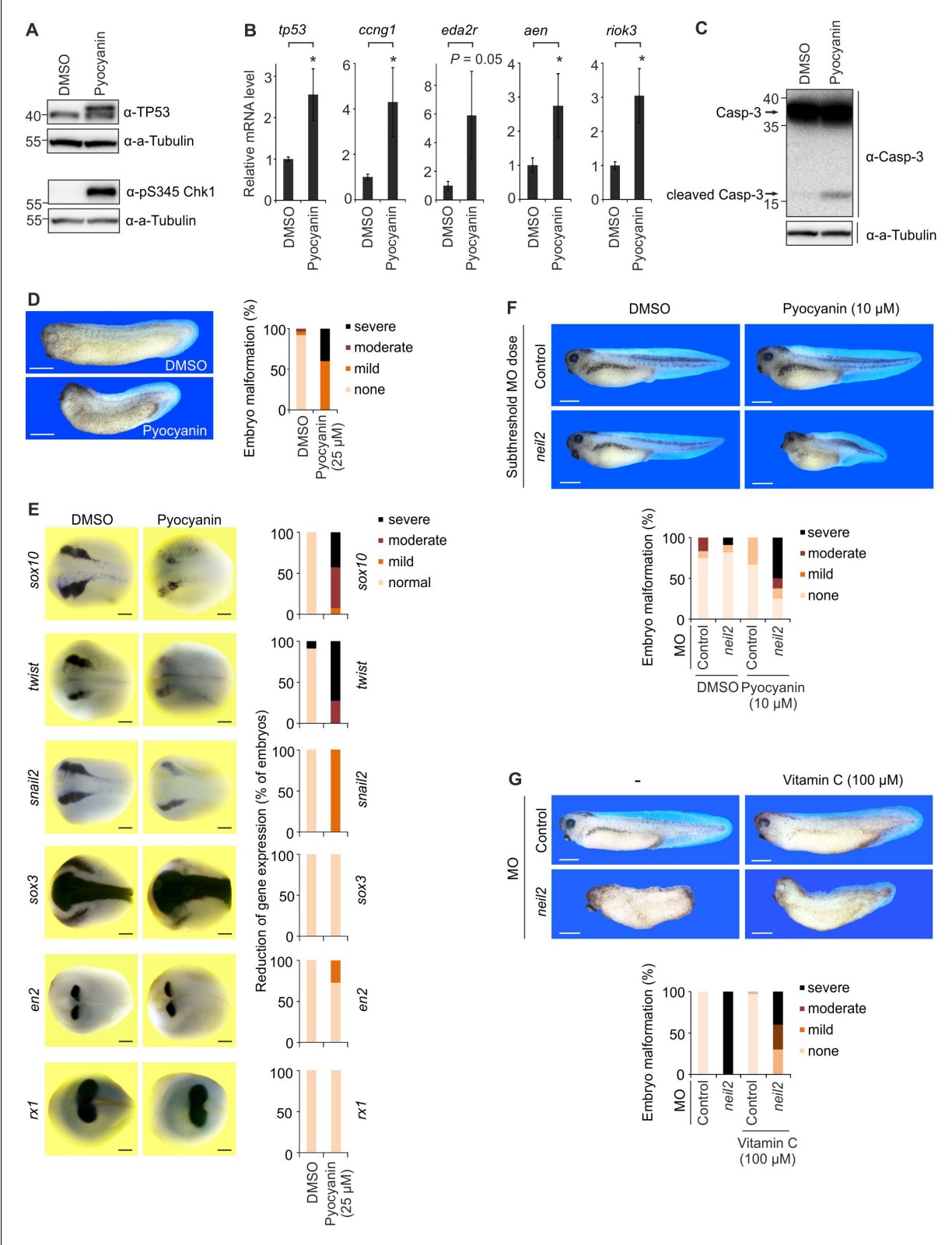

**Figure 3.** Oxidative stress causes neural crest defects in *Neil2*-deficient *Xenopus* embryos. (**A**) Western blot analysis for total Tp53 and pS345 Chk1 of stage 14 embryos cultivated in 25 µM pyocyanin or DMSO (solvent). Alpha (a-) Tubulin served as loading control. Molecular weight of marker proteins [x10$^{-3}$] is indicated on the left. (**B**) qPCR expression analysis of *tp53* and Tp53 target genes in embryos at stage 14 cultivated in 25 µM pyocyanin or DMSO. Expression of examined genes was normalized to *h4* and is presented relative to mRNA level of DMSO treated embryos (mean ±s.d., n = 3

*Figure 3 continued on next page*

*Figure 3 continued*
embryo batches consisting of 6 embryos each). (C) Western blot analysis for Caspase-3 of stage 14 embryos cultivated in 25 µM pyocyanin or DMSO (solvent). Alpha (a-) Tubulin served as loading control. Uncleaved and cleaved (active) Caspase-3 are indicated (arrows). (D) Left, representative phenotypes of stage 32 embryos treated with 25 µM pyocyanin or DMSO (solvent). Right, quantification of embryo malformations (n = 25 and 15 embryos per group, from left to right). Scale bars, 500 µM. (E) Left, whole mount in situ hybridization of the indicated marker genes in stage 16 *Xenopus* embryos treated with 25 µM pyocyanin or DMSO (solvent). Right, quantification of embryo malformations [n = 10 and 14 embryos per group for *sox10*; 2 × 11 (*twist*); 15 and 7 (*snail2*); 11 and 14 (*sox3*); 2 × 11 (*en2*); 15 and 20 (*rx1*), from left to right]. Scale bars, 200 µM (F) Top, phenotypes of stage 39 embryos treated with 10 µM pyocyanin and injected with 15 ng/embryo of control and *neil2* MOs (subthreshold dose). Bottom, quantification of embryo malformations (n = 12, 11, 6 and 8 embryos per group, from left to right). Scale bars, 500 µM. (G) Top, phenotypes of stage 37 embryos injected with 40 ng/embryo of control and *neil2* MOs and treated with 100 µM Vitamin C. Bottom, quantification of embryo malformations (n = 16, 22, 33 and 10 embryos per group, from left to right). Scale bars, 500 µM.
DOI: https://doi.org/10.7554/eLife.49044.005

DDR using pyocyanin, which induced strong *ccng1* expression (*Figure 4B*). As observed in Neil2 morphants, *ccng1* expression was spatially restricted to the neural plate. Spatial restriction of the DDR may be related to patterned expression of *tp53* itself, as in situ hybridization showed preferential *tp53* expression in the neural plate, notably in the anterior wherefrom cNCCs arise (*Figure 4C*). High-level expression of *tp53* in the embryonic CNS is observed in diverse vertebrates, including zebrafish, *Xenopus*, chick, and mouse (*Hoever et al., 1997*; *Lee et al., 2008*; *Rinon et al., 2011*).

The results suggest that the cNCC defects in Neil2 morphants reflect a neural plate-restricted Tp53-response to oxidative DNA damage. Consistently, *tp53* mRNA injection downregulated the cNCC marker *snail2* at mid neurula stage (*Figure 4D*) and induced head and tail abnormalities (*Figure 4E*). Importantly, injection of a *tp53* antisense MO (*Takebayashi-Suzuki, 2003*) not only blocked induction of Tp53 target genes in Neil2 morphants but also rescued phenotypic abnormalities substantially (*Figure 4F–G*). In sum, we propose that Neil2 protects against an oxidative DNA damage-induced Tp53 response and intrinsic apoptosis in the neural plate, thereby safeguarding cNCC development (*Figure 4H*).

## Apex1-deficiency phenocopies neural crest defects of Neil2 morphant *Xenopus* embryos

We asked if other BER factors have roles similar to Neil2 in *Xenopus* embryogenesis. APEX1 (Apurinic/Apyrimidinic Endodeoxyribonuclease 1) functions downstream of DNA glycosylases, processing the abasic (apurinic/apyrimidinic (AP)) sites produced during BER (*Abbotts and Madhusudan, 2010*), and *Apex1*-deficiency in mice leads to early embryonic lethality (*Xanthoudakis et al., 1996*). Injection of *apex1* MO at similar dosage as *neil2* MO induced severe abnormalities and early lethality confirming essentiality of Apex1 also for *Xenopus* embryonic development (data not shown). Interestingly, injection of reduced amounts of *apex1* MO phenocopied Neil2 morphants, with embryos displaying microcephaly, and reduced or absent dorsal and tail fins (*Figure 5A*). Human *APEX1* mRNA partially rescued the phenotype confirming specificity of the *apex1* MO (*Figure 5A*). While Tp53 and phospho-Chk1 levels were unaltered (*Figure 5B*), Tp53 target genes were induced in Apex1 morphants (*Figure 5C*). As in Neil2 morphants, phospho-histone H3 levels were unchanged and Caspase-3 cleavage was induced (*Figure 5D–E*). Hence, the BER enzymes Neil2 and Apex1 exhibit similar functions in *Xenopus* neural crest development.

## *Neil1,2,3* triple-mutant teratomas display cNCC differentiation defects

We next investigated if the role of NEIL DNA glycosylases to protect against ROS damage and safeguard neural crest development is conserved in mammals, and used mouse embryonic stem cells (mESCs) as a model system. We generated *Neil1,2,3* triple-knockout (TKO) mESCs by CRISPR/Cas9 genome editing (*Cong et al., 2013*). We included NEIL3 in addition to NEIL1 and NEIL2 to account for any possible functional redundancy among the NEIL family. The biochemical and biological properties of NEIL3, however, are quite distinct from those of NEIL1 and NEIL2 (*Krokeide et al., 2013*; *Liu et al., 2013*). We flanked and deleted the coding region of the catalytic domains with two gRNAs for each *Neil* gene (*Figure 6—figure supplement 1A*), as validated by genotyping PCR (*Figure 6—figure supplement 1B*). Gene inactivation was further confirmed by western blot analysis for NEIL1 and RT-qPCR for *Neil2* and *Neil3* (*Figure 6—figure supplement 1C-D*). Expression of

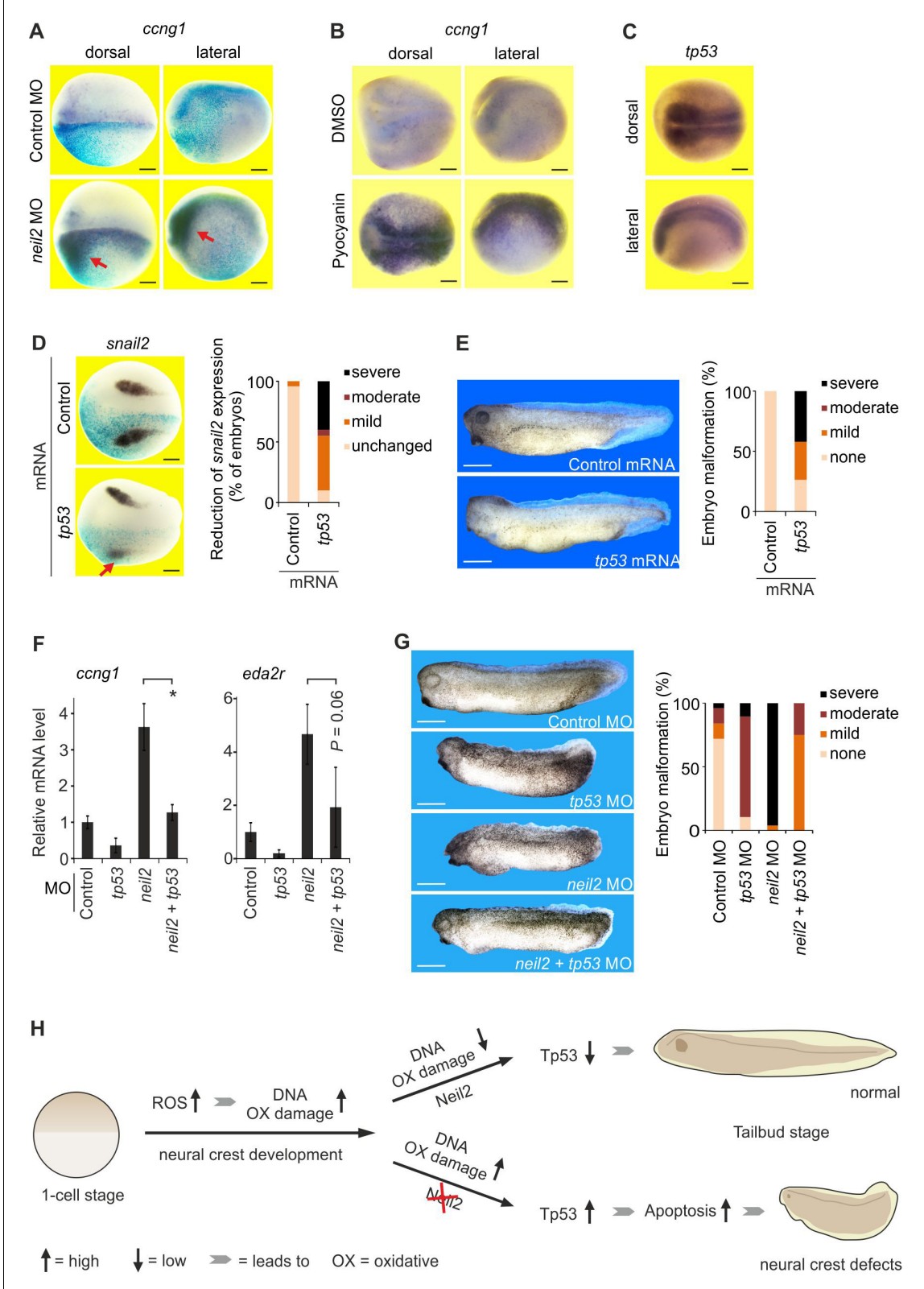

**Figure 4.** A neuroectoderm-restricted Tp53 DNA damage response triggers neural crest defects in *Xenopus* embryos. (A–D) Whole mount in situ hybridization for the indicated marker genes in stage 18 *Xenopus* embryos shown in dorsal view or as indicated. Scale bars, 200 μm. (A) Embryos were injected unilaterally with control or *neil2* MO (lineage-traced by co-injected *lacZ* mRNA, light blue speckles). (B) Embryos were treated with 25 μM pyocyanin or DMSO (solvent). (C) Expression of *tp53* in untreated embryos. (D) Left, embryos were unilaterally injected with control or *tp53* mRNA

*Figure 4 continued on next page*

*Figure 4 continued*

(lineage-traced by co-injected *lacZ* mRNA, light blue speckles). Note reduced *snail2* expression in neural crest cells after *tp53* mRNA injection (red arrow). Right, quantification of reduced *snail2* expression (n = 24 and 20 embryos per group, from left to right). (E) Left, phenotype of stage 32 embryos injected with control or *tp53* mRNA. Right, quantification of embryo malformations (n = 23 and 19 embryos per group, from left to right). Scale bars, 500 µm. (F) qPCR expression analysis of *ccng1* and *eda2r* in embryos at stage 14 injected with MOs as indicated. Expression of *ccng1* and *eda2r* was normalized to *h4* expression and is presented relative to control MO-injected embryos. (mean ±s.d., n = 3 embryo batches consisting of 6 embryos each). (G) Left, phenotypes of stage 32 embryos injected with the indicated MOs. Scale bars, 500 µm. Right, quantification of embryo malformations (n = 25, 26, 19 and 28 embryos per group, from left to right). (H) Model for Neil2 function in *Xenopus* neural crest specification. During neural crest development ROS levels are increased and DNA is oxidatively damaged. Unrepaired DNA damage in the absence of Neil2 induces Tp53-DDR followed by intrinsic apoptosis and malformation of neural crest derivatives in the developing embryo. ROS, reactive oxygen species.

DOI: https://doi.org/10.7554/eLife.49044.006

pluripotency markers was unaltered for *Pou5f1* and *Klf4*, whereas *Nanog* expression was slightly reduced in *Neil1,2,3* TKO mESCs compared to control cells that originated from mock transfections lacking specific guide RNAs (*Figure 6—figure supplement 1E*).

We subjected control and *Neil*-TKO mESCs to teratoma assays (*Ralston and Rossant, 2010*) using three independent clones of each, to average-out clonal variation. When transplanted subcutaneously into immunodeficient mice, teratomas grew from all six mESC lines. Histological analysis of control and *Neil*-TKO teratomas revealed derivatives of ectoderm, endoderm and mesoderm in all samples confirming pluripotency of *Neil*-deficient mESCs (*Figure 6A*). However, transcriptome analysis by RNA-seq uncovered thousands of genes differentially expressed between control and *Neil*-TKO teratomas (*Figure 6B*). Intriguingly, pathway enrichment analysis of >2 fold differentially expressed genes yielded one significant hit for the up- and downregulated genes each, 'PluriNet-Work' and 'neural crest differentiation', respectively (*Figure 6C*). The term 'PluriNetWork' refers to the genes regulating pluripotency in mouse stem cells (*Som et al., 2010*). We confirmed upregulation of pluripotency markers (*Pou5f1, Nanog and Klf4*), suggesting incomplete silencing of the pluripotent state in *Neil*-TKO teratomas (*Figure 6D*). Downregulated genes included neural crest effectors such as *Pax3* (*LaBonne and Bronner-Fraser, 1998*), *Tfap2b, Phox2b, Dbh, Crabp1, Neurog1* and *Wnt3a*, besides a suite of downregulated *Hox* genes (*Hoxa2, Hoxa3, Hoxa4, Hoxa5, Hoxa9, Hoxb1, Hoxb2, Hoxb3, Hoxb4, Hoxb5, Hoxc4, Hoxc5, Hoxd3*), which are prominently expressed during neural crest/pharyngeal arch patterning, where they control head skeletal development (*Trainor and Krumlauf, 2001*) (*Supplementary file 2*). In fact, marker gene analysis for endoderm (*Gata6*), mesoderm (*Eomes*), neuroectoderm (*Pax6, Nestin, Sox1 and Pax2*), and neural crest (*Pax3, Hoxa2, Tfap2b and Neurog1*) indicated mild neural and severe neural crest differentiation defects (*Figure 6E* and *Figure 6—figure supplement 2*). Moreover, the TP53 target genes *Ccng1, Mdm2, Sesn2* and *Eda2r* were significantly upregulated in *Neil* TKO teratomas (*Figure 6F*) indicative of a TP53 DDR. These results indicate that the requirement for NEIL function in cNCC development is evolutionarily conserved between amphibians and mammals.

## Neural and cNCC differentiation defects are caused by *Neil1*- and *Neil2*-deficiency

To analyze the individual requirement of NEILs for neural and cNCC specification, we generated single *Neil1*, −2, *and* −3 mutant mESCs (*Figure 7—figure supplement 1A–D*). *Neil*-mutant mESCs were subjected to in vitro differentiation for eight days in embryoid bodies (EB) in absence or presence of retinoic acid (RA), the latter favoring neural differentiation (*Bibel et al., 2007*). *Neil3* mutants were largely unaffected for all markers tested (*Figure 7A*). In contrast, single *Neil1* and *Neil2* mutants showed significant reduction in neural crest marker (*Pax3, Hoxa2, Tfapb2, Neurog1*) and also pan-neuroectodermal marker (*Pax6, Nestin, Sox1, Pax2*) expression, while endoderm- and mesoderm markers were unaffected (*Figure 7A* and *Figure 7—figure supplement 2*). Importantly, *Pax3* and *Pax6* induction during differentiation was partially restored in *Neil1*- and *Neil2*-deficient cells by stable transfection with catalytically active- but not inactive human NEIL1- and NEIL2-encoding constructs (*Figure 7—figure supplement 3A*). Note that the degree of rescue was likely limited by the low expression of the transfected NEIL constructs (*Figure 7—figure supplement 3B*). This rescue not only confirms specificity of the *Neil* knockout approach but also demonstrates that the neural and cNCC differentiation defects are not due to mutant mESCs having undergone irreversible

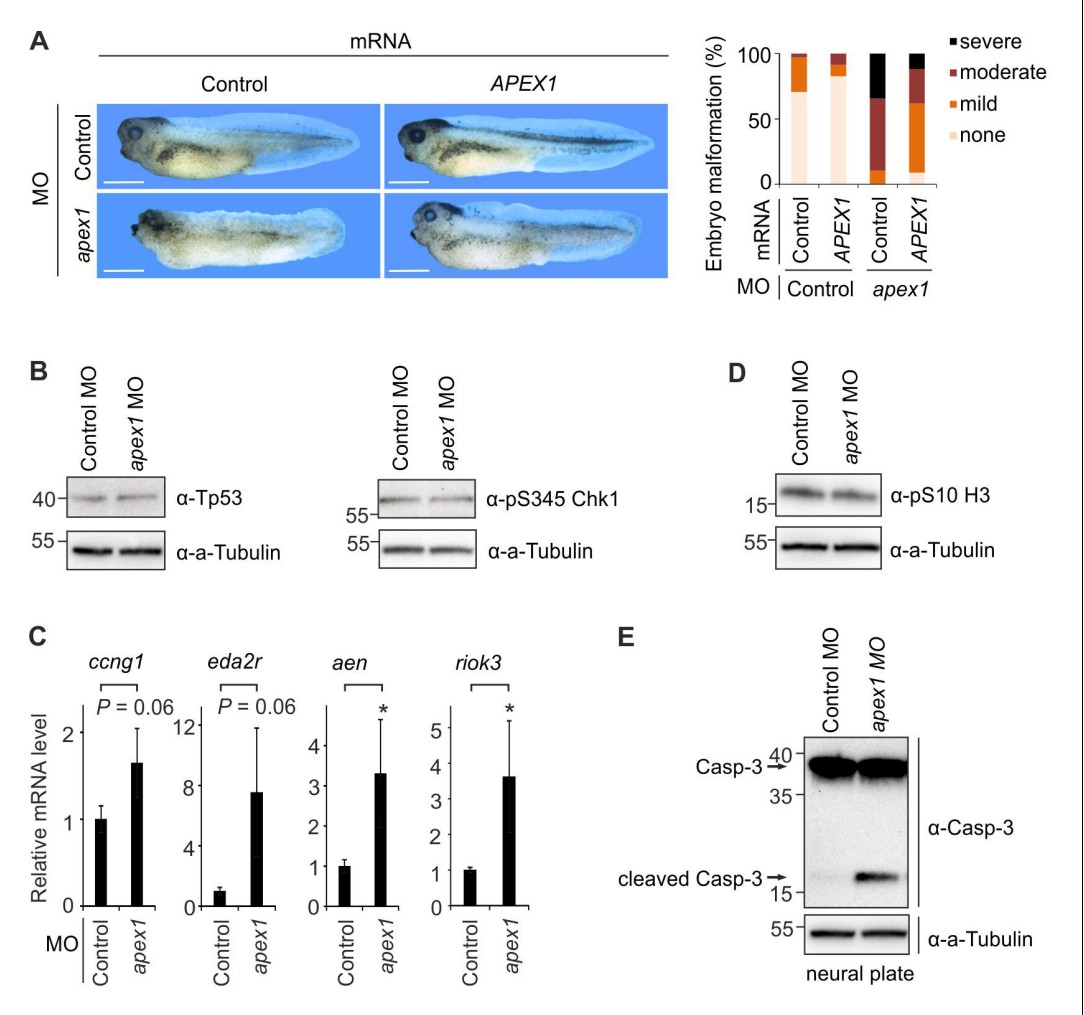

**Figure 5.** *Apex1*-deficiency phenocopies neural crest defects of *Neil2*-morphant *Xenopus* embryos. (**A**) Left, phenotypes of stage 39 embryos injected with control and *apex1* MO, and control and human *APEX1* mRNA for phenotypic rescue. Scale bars, 500 μm. Right, quantification of embryo malformation in all four injection groups (n = 34, 23, 29 and 34 embryos per group, from left to right). (**B**) Western blot for total Tp53 and phosphoserine (pS345) Chk1 in stage 14 control and *apex1* MO-injected embryos. Alpha (a-) Tubulin served as loading control. Molecular weight of marker proteins [x10$^{-3}$] is indicated on the left. (**C**) qPCR expression analysis of Tp53 target genes in control and *apex1* MO-injected embryos at stage 14. Expression of target genes was normalized to *h4* and is presented as relative mRNA level of control MO-injected embryos (mean ±s.d., n = 3 embryo batches consisting of 6 embryos each). (**D**) Western blot for phosphoserine (pS10) histone H3 from control and *apex1* MO-injected stage 14 embryos. Alpha (a-) Tubulin served as loading control. (**E**) Western blot analysis for Caspase-3 in control and *apex1* MO-injected stage 14 embryos. Alpha (a-) Tubulin served as loading control. Uncleaved and cleaved (active) Caspase-3 are indicated (arrows).
DOI: https://doi.org/10.7554/eLife.49044.007

changes/DNA damage. Instead, the rescue indicates an acute requirement for NEIL1 and NEIL2 during cNCC differentiation.

To corroborate neural and cNCC developmental defects, we carried out RNA-seq analysis of undifferentiated- and embryoid-body differentiated *Neil1* and *Neil2* single-mutant mESCs. While we observed hundreds of differentially expressed (DE) genes in mutant *Neil1* and *Neil2* mESCs (*Figure 7B* and *Supplementary file 3–4*), they did not cluster when subjected to pathway enrichment analysis. Upon differentiation, the number of DE genes increased in *Neil1*- and *Neil2*-deficient cells substantially to several thousand, notably in EBs treated with RA, with up- and downregulated genes equally distributed (*Figure 7B* and *Supplementary file 3–4*). The majority of DE genes

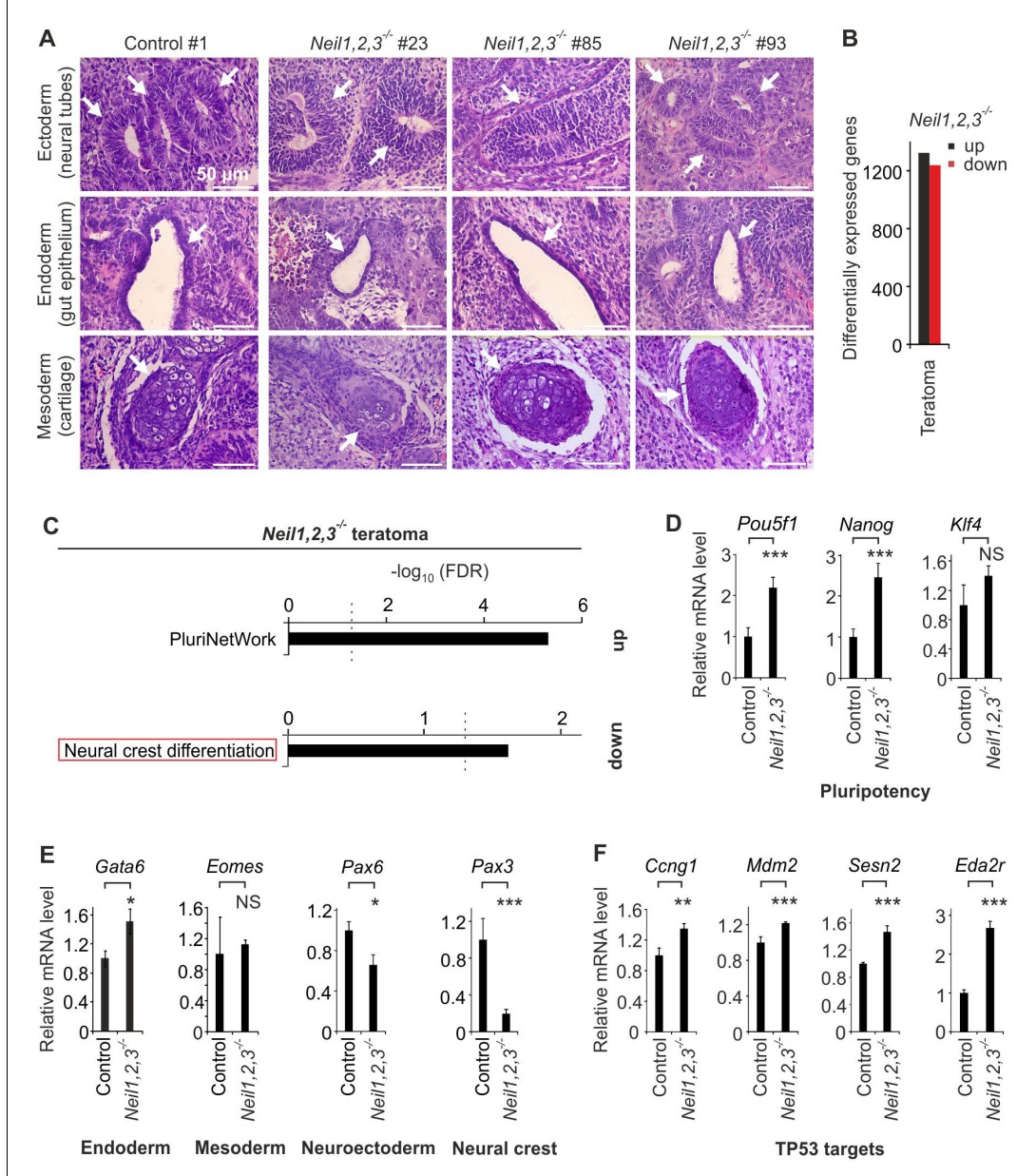

**Figure 6.** *Neil1,2,3*-deficient mESCs display neural crest cell differentiation defects. (A) Histological analysis of H and E stained teratomas derived from control and *Neil1,2,3*-deficient mESC lines. Arrows indicate neural tube- (ectoderm), gut epithelium- (endoderm) and cartilage- (mesoderm) related structures within each section. (B) Quantification of differentially expressed genes in *Neil1,2,3*-deficient teratomas. (C) Pathway enrichment analysis of up- and downregulated genes in *Neil1,2,3*-deficient teratomas. Dashed line indicates the significance threshold FDR = 0.05. (D) qPCR expression analysis of pluripotency genes in control and *Neil1,2,3* triple-deficient teratomas. Marker gene expression was normalized to *Tbp* and is presented relative to control teraomas (mean ±s.d., n = 3 biological replicates with each three technical replicates). (E) qPCR expression analysis as in (D) but of endoderm (*Gata6*), mesoderm (*Eomes*), neuroectoderm (*Pax6*) and neural crest (*Pax3*) marker genes of control and *Neil1,2,3*-deficient teratomas. (F) qPCR expression analysis as in (D) but of selected TP53 target genes.

DOI: https://doi.org/10.7554/eLife.49044.008

The following figure supplements are available for figure 6:

**Figure supplement 1.** Generation and characterization of *Neil*-deficient mESCs.
DOI: https://doi.org/10.7554/eLife.49044.009
**Figure supplement 2.** Extended neural and neural crest marker gene analysis in *Neil* triple-deficient teratomas.
DOI: https://doi.org/10.7554/eLife.49044.010

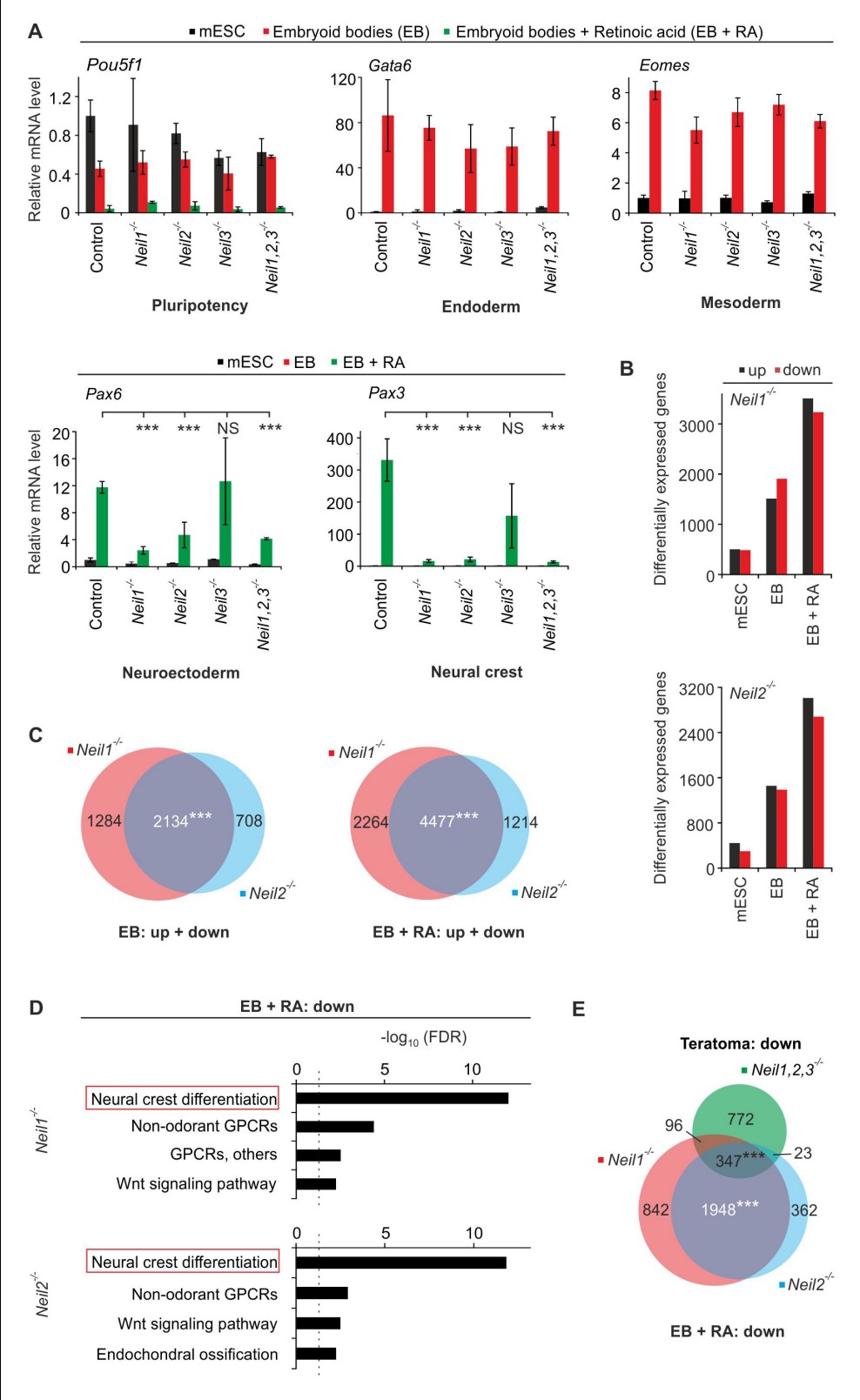

**Figure 7.** *Neil1*- and *Neil2*-deficient mESCs display neural crest cell differentiation defects in vitro. (**A**) qPCR expression analysis of pluripotency (*Pou5f1*), endoderm (*Gata6*), mesoderm (*Eomes*), neuroectoderm (*Pax6*) and neural crest (*Pax3*) marker genes of the indicated *Neil*-mutant mESCs. Cells were differentiated into embryoid bodies (EBs) without or with retinoic acid (EBs + RA). Expression of marker genes was normalized to *Tbp* and is
*Figure 7 continued on next page*

*Figure 7 continued*

relative to control clones in mESC state. (s.d., n = 3 biological replicates). (B) Quantification of differentially expressed genes in *Neil1* and *Neil2* single-deficient mESCs, EBs and EBs + RA. (C) Overlap of differentially expressed genes from *Neil1* and *Neil2* single-deficient EBs (left panel) and EBs + RA (right panel) (D) Pathway enrichment analysis of downregulated genes from *Neil1* (upper) and *Neil2* (lower panel) single-deficient EBs + RA. Dashed lines indicate the significance threshold FDR = 0.05. (E) Overlap of downregulated genes from *Neil1* and *Neil2* single-deficient EBs + RA, and downregulated genes from *Neil1,2,3*-deficient teratomas.
DOI: https://doi.org/10.7554/eLife.49044.011
The following figure supplements are available for figure 7:

**Figure supplement 1.** Characterization of *Neil* single-deficient mESCs.
DOI: https://doi.org/10.7554/eLife.49044.012
**Figure supplement 2.** Extended neural and neural crest marker gene analysis in *Neil* single-deficient EBs + RA.
DOI: https://doi.org/10.7554/eLife.49044.013
**Figure supplement 3.** Knockout specificity of *Neil*-deficient cells.
DOI: https://doi.org/10.7554/eLife.49044.014
**Figure supplement 4.** Gene misregulation in *Neil*-deficient embryoid bodies.
DOI: https://doi.org/10.7554/eLife.49044.015

---

overlapped between *Neil1* and −2 mutant EBs and EBs + RA (*Figure 7C*), supporting functional commonality between NEIL1 and NEIL2. Importantly, pathway enrichment analysis of the downregulated genes revealed 'neural crest differentiation' as the top hit in both genotypes and in both differentiation regimes (*Figure 7D* and *Figure 7—figure supplement 4A*). Downregulated genes included the neural crest effectors *Pax3*, *Tfap2b*, *Phox2b*, *Crabp1*, *Neurog1* and a series of *Hox* genes (*Supplementary file 3–4*) similarly as for *Neil*-TKO teratomas. Downregulated genes from *Neil*-TKO teratomas significantly overlapped with downregulated genes from either *Neil1*- or *Neil2*-mutant EBs and EBs + RA (*Figure 7—figure supplement 4B* and *Figure 7E*). We conclude that in vitro differentiation of *Neil1*- and *Neil2* single-mutant mESCs recapitulates neural and cNCC differentiation defects observed in *Neil*-TKO teratomas. Besides, the upregulated genes of both *Neil1*- and *Neil2*-deficient EBs + RA were significantly enriched for the pathway term 'TYROBP causal network' (*Figure 7—figure supplement 4C*), associated with late-onset Alzheimer's disease (*Zhang et al., 2013*).

## *Apex1*-deficiency phenocopies neural and cNCC differentiation defects of *Neil1* and *Neil2* mutants

We asked if *Apex1*-deficiency in mESC differentiation phenocopies *Neil*-deficiency as observed in *Xenopus* embryos. Hence, we generated and validated an *Apex1* mESC knockout-line (*Apex1* #46, *Figure 8—figure supplement 1A–D*) and subjected it to in vitro differentiation. Expression of germ layer marker genes was reduced for all tested tissues in the *Apex1*-mutant line (*Figure 8A*), indicating a more severe differentiation defect than in *Neil1* and *Neil2* mutants. Yet, RNA-seq analysis revealed a substantial overlap of commonly deregulated genes between *Neil1*-, *Neil2*- and *Apex1*-deficient EBs treated with RA (*Figure 8B–C* and *Supplementary file 5*). Moreover, pathway enrichment analysis of the downregulated genes in *Apex1* EBs + RA once again resulted in 'neural crest development' as the top hit (*Figure 8D*). Thus, *Apex1*-deficient mESCs substantially phenocopy cNCC differentiation defects of *Neil* mutants similar to *Xenopus* embryos.

## Neural and cNCC differentiation in embryoid bodies is independent of oxidative DNA demethylation

We tested if active removal of the oxidative demethylation intermediates 5fC and 5caC is required for neural and cNCC differentiation in EBs. To this end, we generated a *Tdg* mESC knockout-line using CRISPR/Cas9 (*Tdg* #25, *Figure 9—figure supplement 1A–C*). *Tdg* knockout mice are embryonic lethal and *Tdg*-deficient mESCs fail to undergo terminal neuronal differentiation (*Cortázar et al., 2011*). As expected, *Tdg*-deficient mESCs had 3–4-fold increased genomic 5fC and 5caC levels (*Shen et al., 2013*; *Steinacher et al., 2019*), but they showed no change in pluripotency marker expression (*Figure 9—figure supplement 1D–E*). Moreover, *Tdg*-deficient cells subjected to EB differentiation induced marker gene expression of all germ layers with no significant difference

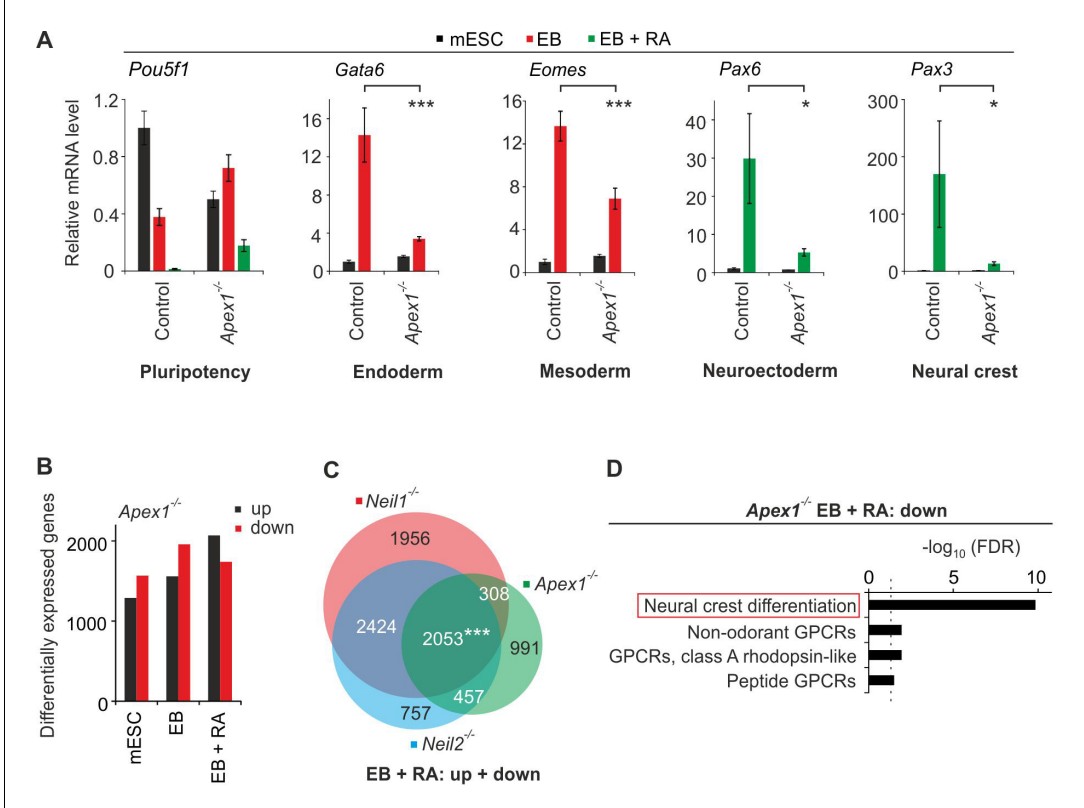

**Figure 8.** *Apex1*-deficiency leads to neural crest cell differentiation defects. (**A**) qPCR expression analysis of pluripotency (*Pou5f1*), endoderm (*Gata6*), mesoderm (*Eomes*), neuroectoderm (*Pax6*) and neural crest (*Pax3*) marker genes of control and *Apex1*-deficient mESCs, EBs and EBs + RA. Expression of marker genes was normalized to *Tbp* and is relative to control mESCs (s.d., n = 3 technical replicates). (**B**) Quantification of differentially expressed genes in *Apex1*-deficient mESCs, EBs and EBs + RA. (**C**) Overlap of differentially expressed genes from *Neil1*, *Neil2* and *Apex1* single-deficient EBs + RA. (**D**) Pathway enrichment analysis of downregulated genes from *Apex1*-deficient EBs + RA. Dashed line indicates the significance threshold FDR = 0.05.

DOI: https://doi.org/10.7554/eLife.49044.016

The following figure supplement is available for figure 8:

**Figure supplement 1.** Construction and characterization of *Apex1*-deficient mESCs.

DOI: https://doi.org/10.7554/eLife.49044.017

from control cells (*Figure 9A*). We conclude that oxidative *Tdg*-dependent DNA demethylation is not required for early cNCC differentiation in embryoid bodies.

NEIL1 and NEIL2 are involved in handover and processing of abasic sites during oxidative DNA demethylation after 5fC/5caC excision by TDG (*Schomacher et al., 2016*). Since abasic sites are genotoxic, *Neil*-deficiency may trigger a DDR because of accumulation of unprocessed TET/TDG-demethylation intermediates. If so, preventing 5fC/5caC excision in the first place should rescue the differentiation defects in *Neil*-deficient cells. To block 5fC/5caC excision, we generated a *Tdg*-knock-out mESC line in a *Neil1*-deficient background (*Neil1* #7/*Tdg* #11, *Figure 9—figure supplement 2A–E*). However, in the *Neil1/Tdg* double-knockout line there was no rescue of cNCC differentiation, while the *Tdg*-single mutant line expectedly showed normal neural and cNCC marker gene expression (*Figure 9B*). We conclude that the differentiation defects induced by deficiency of *Neils* are independent of a role in oxidative DNA demethylation.

## Mitochondrial oxidative DNA damage is increased in *Neil1*- and *Neil2*-deficient cells during neural differentiation

To test if defective neural and cNCC differentiation is related to elevated oxidative DNA damage as in *Xenopus*, we differentiated mESCs in the presence of pyocyanin. Consistently, pyocyanin inhibited neural and cNCC gene expression upon RA-induced EB differentiation, without affecting endoderm

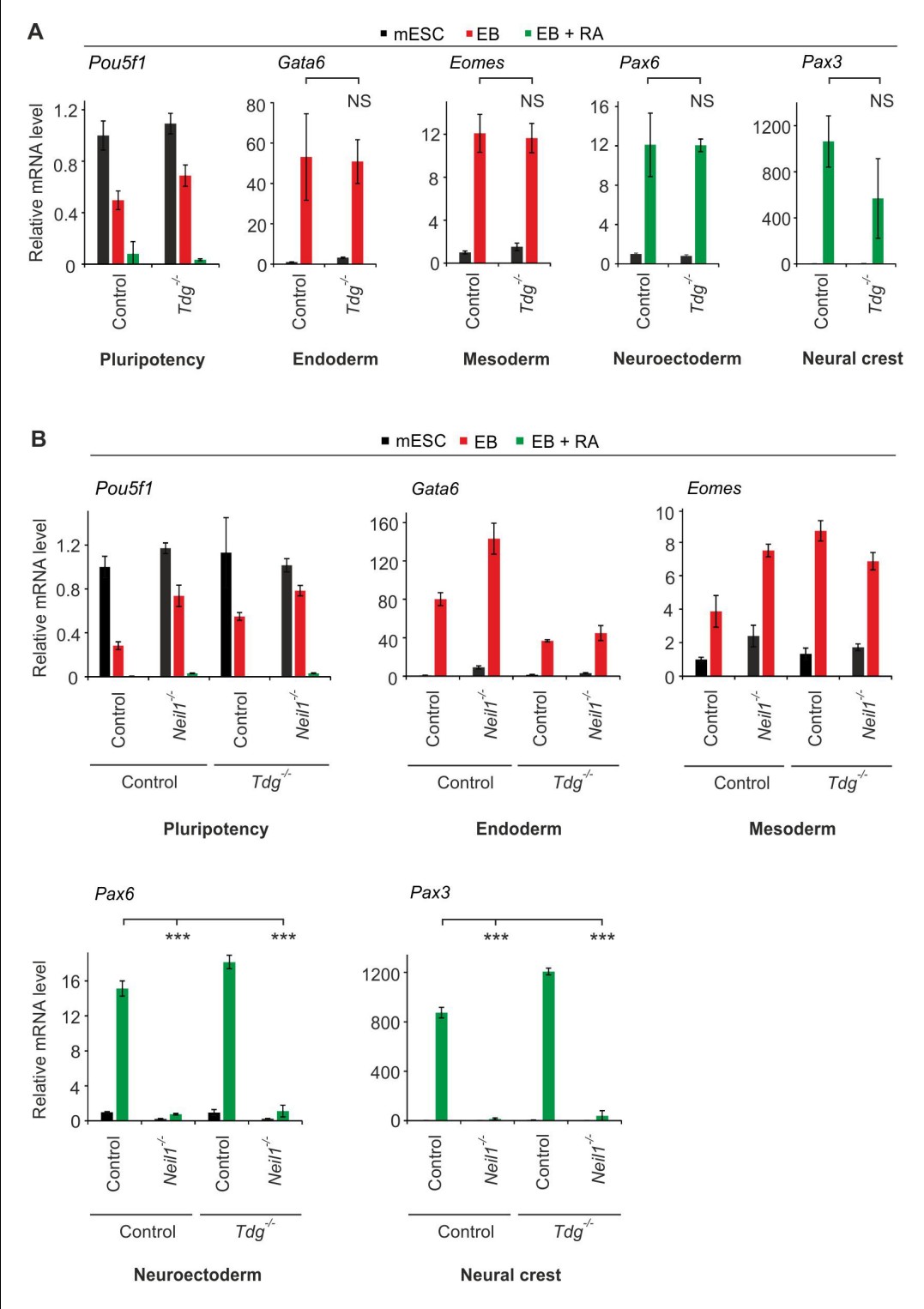

**Figure 9.** Neural differentiation is independent of the role of NEIL DNA glycosylases in oxidative DNA demethylation. (**A**) qPCR expression analysis of pluripotency (*Pou5f1*), endoderm (*Gata6*), mesoderm (*Eomes*), neuroectoderm (*Pax6*) and neural crest (*Pax3*) marker genes of control and *Tdg*-deficient mESCs, EBs, and EBs + RA. Expression of marker genes was normalized to *Tbp* and is relative to control mESCs (s.d., n = 3 technical replicates). (**B**) qPCR expression analysis as in (**A**) but of control, *Neil1* and *Tdg* single- and double-deficient mESCs, EBs and EBs + RA. Expression of marker genes was normalized to *Tbp* and is relative to double-control clone in mESC state (s.d., n = 3 technical replicates).

*Figure 9 continued on next page*

*Figure 9 continued*

DOI: https://doi.org/10.7554/eLife.49044.018

The following figure supplements are available for figure 9:

**Figure supplement 1.** Construction and characterization of *TDG*-deficient mESCs.
DOI: https://doi.org/10.7554/eLife.49044.019

**Figure supplement 2.** Construction and characterization of *Neil1/TDG* double-deficient mESCs.
DOI: https://doi.org/10.7554/eLife.49044.020

or mesoderm differentiation (*Figure 10A*). In addition, upon differentiation, TP53 target gene expression significantly increased in presence of pyocyanin compared to mock treatment (*Figure 10B*). The results align with the observation that oxidative stress impairs cNCC differentiation (*Chen and Sulik, 1996*; *Sakai and Trainor, 2016*; *Yan et al., 2010*).

Hence, the *Xenopus* and mESC data converge on the conclusion that NEIL1 and NEIL2 are required to repair oxidative base lesions during early neural development. Since NEIL1 and NEIL2 localize to- and maintain genomic stability in the nucleus as well as in mitochondria (*Hu et al., 2005*; *Mandal et al., 2012*; *Prakash and Doublié, 2015*; *Vartanian et al., 2006*), this raised the question, in which of these two compartments NEILs may be required during early embryogenesis. To quantify NEIL-processed lesions, we developed a novel protocol. We isolated gDNA and mtDNA (>14 fold

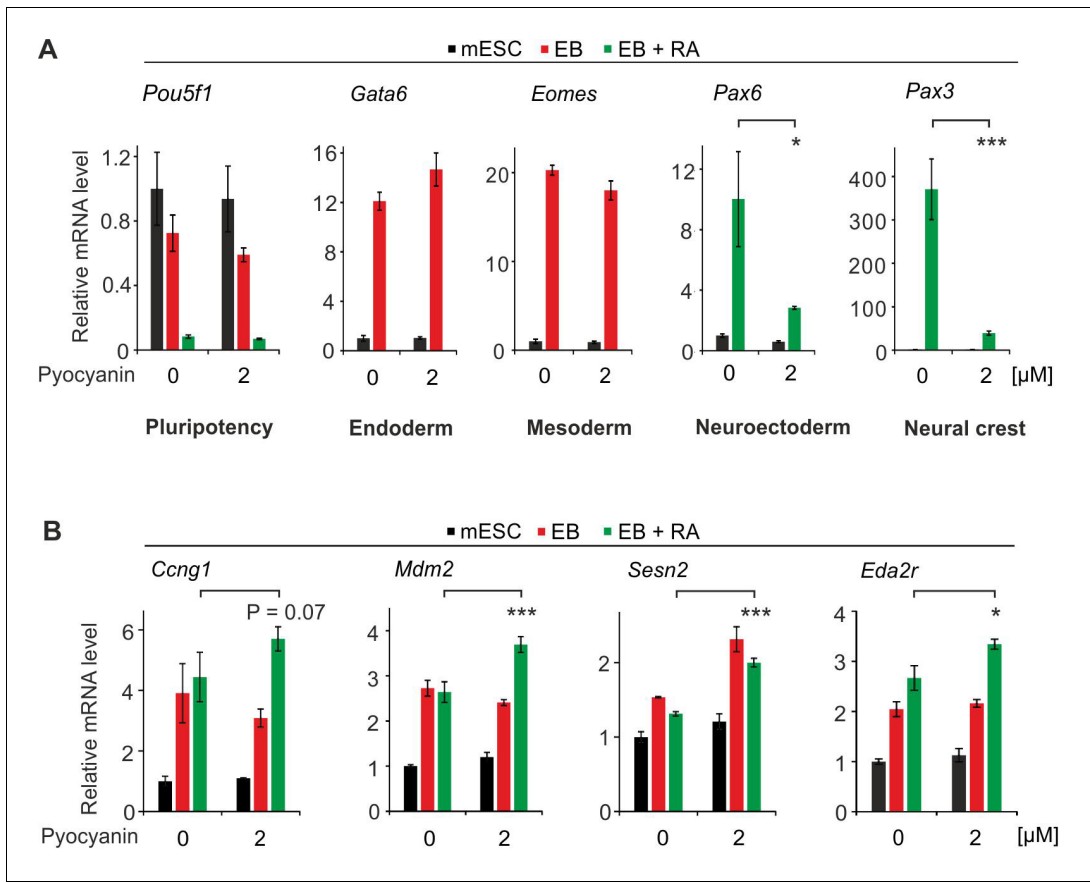

**Figure 10.** Oxidative stress impairs cNCC differentiation in mESCs. (**A**) qPCR expression analysis of pluripotency (*Pou5f1*), endoderm (*Gata6*), mesoderm (*Eomes*), neuroectoderm (*Pax6*) and neural crest (*Pax3*) marker genes of mock and pyocyanin-treated control mESCs, EBs and EBs + RA. Expression of marker genes was normalized to *Tbp* and is relative to mock-treated mESCs (s.d., n = 3 technical replicates). (**B**) qPCR expression analysis as in (**A**) but of selected TP53 target genes of mock (DMSO) and pyocyanin-treated control mESCs, EBs and EBs + RA.
DOI: https://doi.org/10.7554/eLife.49044.021

enriched for mtDNA, *Figure 11—figure supplement 1A*), and digested it with *E. coli* EndoIII, a bifunctional DNA glycosylase/AP lyase that excises a similar spectrum of oxidatively damaged DNA bases as NEIL1 and NEIL2 (*Dizdaroglu et al., 2000*). EndoIII base excision at oxidative lesions generates abasic sites (*Hatahet et al., 1994*), which are then quantified by LC-MS/MS mass spectrometry (*Rahimoff et al., 2017*). Since steady state levels of endogenous abasic sites in DNA are more abundant than oxidative base damages (*Swenberg et al., 2011*) we pretreated the DNA with recombinant APEX1 prior to the EndoIII reaction in order to reduce the background from preexisting abasic sites (*Figure 11A* and *Figure 11—figure supplement 1B*). Quantification of abasic sites on synthetic oligonucleotides mixed with known amounts of AP sites accurately matched the expected result (*Figure 11—figure supplement 1C*), validating the method. We noted, though, that APEX1-treatment did not completely erase abasic sites on oligonucleotides under our reaction conditions. Similarly, EndoIII processed the oxidative damage 5hU on oligonucleotides to abasic sites, but also not completely (*Figure 11—figure supplement 1C*).

APEX1-treatment of purified gDNA from mESCs reduced endogenous abasic sites by two-fold, from ~229.000 sites per genome to ~120.000 sites per genome. Subsequent EndoIII incubation led to a statistically significant increase of ~29.000 abasic sites per genome (*Figure 11—figure supplement 1D*). Thus, steady state levels of EndoIII-reactive DNA damage sites were ~8 fold lower than the level of abasic sites, in agreement with previous reports (*Swenberg et al., 2011*). Levels of endogenous abasic sites (without APEX1/EndoIII treatment) were unaffected by *Neil1-* and *Neil2-*deficiency in gDNA and mtDNA, regardless of whether mESCs were differentiated or not (*Figure 11—figure supplement 1E*). Likewise, there was no significant increase in EndoIII-processed sites in gDNA from *Neil1-* and *Neil2-*knockout cells (*Figure 11B*, top). In contrast, in mtDNA from *Neil1-* and *Neil2-*deficient mESCs, EndoIII-created abasic sites were elevated only upon neural differentiation (EB + RA), but not in EBs or undifferentiated mESCs (*Figure 11B*, bottom).

We confirmed elevated mtDNA damage in *Neil1-* and *Neil2-*deficient EBs + RA by an independent method (*Gureev et al., 2017*). Using long-range PCR, levels of DNA damage are monitored based on the fact that DNA lesions inhibit DNA polymerase and slow down accumulation of the PCR product. Therefore, the rate of product amplification is inversely proportional to the number of damaged DNA molecules. Comparing *Neil*-deficient to control cells, this approach revealed increased mtDNA damage in EBs treated with RA, and to lesser extend in EBs without RA (*Figure 11C*).

Moreover, as in *Xenopus* embryos we detected significant upregulation of mitochondrial and nuclear genes encoding components of oxidative phosphorylation as sign of mitochondrial dysfunction in *Neil1-* and *Neil2-*knockout cells, specifically under EB + RA treatment (*Figure 11D*) (*Babenko et al., 2018*; *Heddi et al., 1999*; *Reinecke et al., 2009*). Together, the results suggest that NEIL1 and NEIL2 are specifically required for processing of oxidative lesions occurring in mitochondrial DNA during neural differentiation.

## Mitochondrial TP53 DDR causes neural and cNCC differentiation defects upon *Neil1-* and *Neil2-*deficiency

We asked if *Neil*-deficiency in mESCs elicits a DNA damage response as in *Xenopus* embryos. Indeed, we found a ~ 40% overlap between the 116 top TP53 target genes (*Fischer, 2017*) and the upregulated genes in *Neil1-* and *Neil2-*mutant EBs + RA (*Figure 12A*), but no significant overlap with *Neil1-* and *Neil2-*deficient EBs and mESCs (*Figure 12—figure supplement 1A–B*). Furthermore, differentiation of control mESCs in presence of the TP53 stabilizer NSC 146109 (*Berkson et al., 2005*) resulted in specific neural and cNCC differentiation defects, thus mimicking *Neil*-deficiency (*Figure 12B*). Moreover, we tested if TP53 inhibition could rescue impaired differentiation of *Neil1-* and *Neil2-*mutant mESCs. We differentiated *Neil1* and *Neil2* single mutant mESCs in the presence of Pifithrin-α, an inhibitor of TP53 (*Komarov et al., 1999*). Strikingly, upon Pifithrin-α treatment the neuronal marker *Pax6* and neural crest marker *Pax3* in *Neil1-* and *Neil2-*mutant EBs + RA regained expression levels of control cells, while endoderm and mesoderm differentiation was marginally inhibited in all tested genotypes (*Figure 12C*). In control EBs + RA Pifithrin-α treatment did not affect *Pax6* and *Pax3* expression. We conclude that an upregulated TP53 DDR impairs neural and cNCC differentiation in *Neil1-* and *Neil2-*mutant cells.

Since we observed mtDNA damage accumulation and mitochondrial dysfunction in *Neil1-* and *Neil2-*deficient cells, we tested specifically for a mitochondrial TP53 DDR (*Vaseva and Moll, 2009*) as in *Xenopus*. Expression of the anti-apoptotic factor *Bcl2* was strongly downregulated in *Neil-*

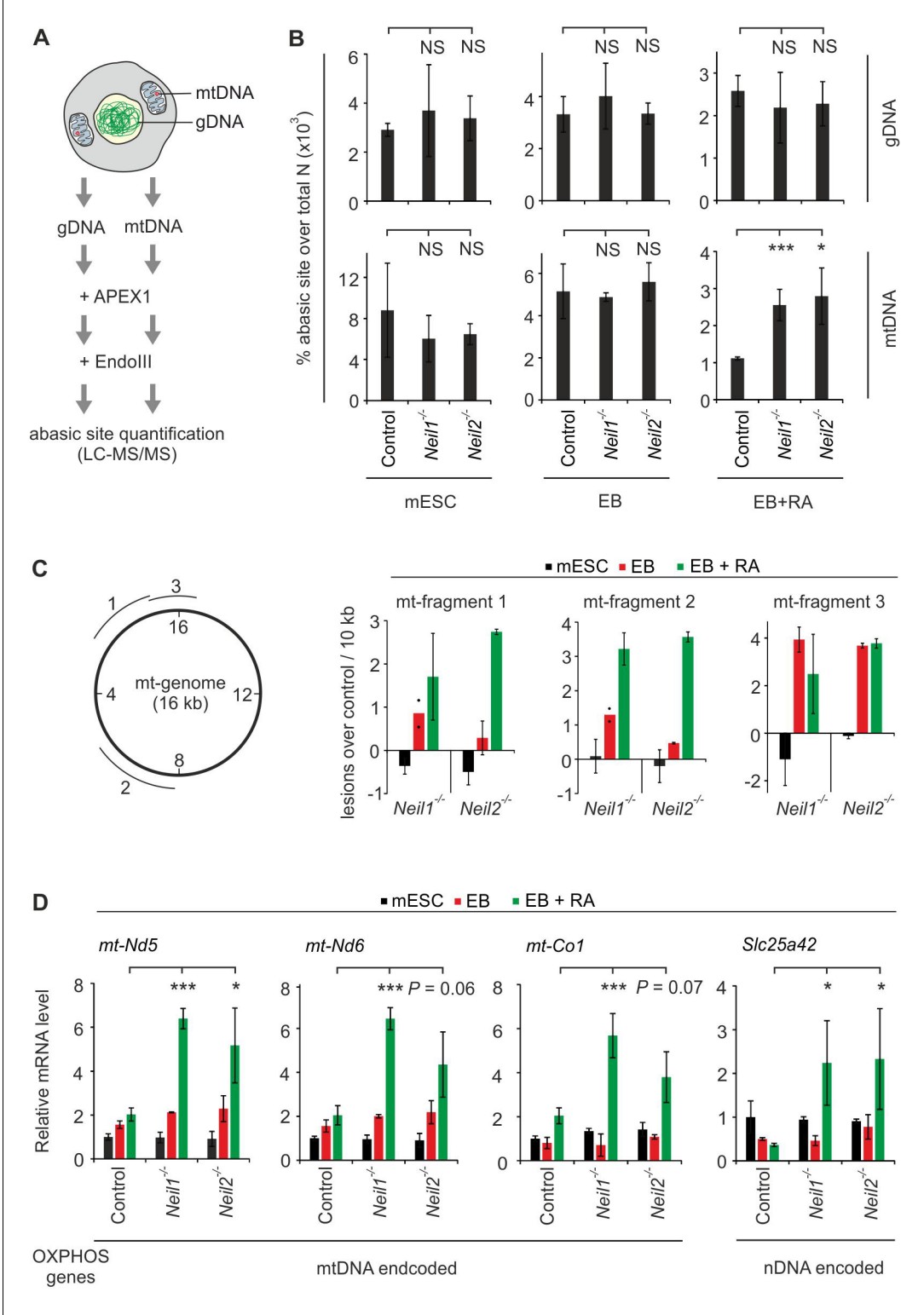

**Figure 11.** Oxidative mtDNA damage and mitochondrial dysfunction in *Neil1-* and *Neil2-*deficient embryoid bodies. (**A**) Workflow to quantify oxidative base lesions in DNA. Isolated genomic and mtDNA is consecutively treated with recombinant APEX1 and EndoIII to monitor endogenous EndoIII-processed oxidative base damages by LC-MS/MS. (**B**) LC-MS/MS quantification of base lesions as described in (**A**) using gDNA and mtDNA of control, *Neil1-* and *Neil2-*deficient mESCs, EBs and EBs + RA as indicated (s.d., n = 3 biological replicates). Abasic site levels are presented in percent of total amount of nucleotides (**N**). (**C**) Left, position of the mitochondrial (mt-)

*Figure 11 continued*

genomic fragments tested for DNA damage (1-3) relative to the nucleotide annotation of the mt genome (inner circle numbers in kb). Right, mtDNA damage in mt-fragments 1–3 in *Neil1-* and *Neil2*-deficient mESCs, EBs and EBs + RA, respectively, calculated as lesions over control in 10 kb (s.d., n = 3 biological replicates). Note, a negative value corresponds to less damage in *Neil*-deficient compared to control cells. (**D**) qPCR expression analysis of genes for oxidative phosphorylation (OXPHOS) encoded either on mitochondrial (mt-) or nuclear (n-) DNA in control, *Neil1* and *Neil2* single-mutant mESCs, EBs and EBs + RA. Expression of marker genes was normalized to *Tbp* and is relative to control mESCs (s.d., n = 3 biological replicates).
DOI: https://doi.org/10.7554/eLife.49044.022
The following figure supplement is available for figure 11:

**Figure supplement 1.** LC-MS/MS analysis of abasic sites in gDNA and mtDNA of *Neil1-* and *Neil2*-deficient cells.
DOI: https://doi.org/10.7554/eLife.49044.023

mutant EBs and EBs + RA, while expression of *Bak1*, a pro-apoptotic factor of the *Bcl2* family (*Graupner et al., 2011*; *Tsujimoto, 1998*), was significantly induced in *Neil*-deficient EBs + RA (*Figure 12D*), consistent with an intrinsic/mitochondrial TP53 response. Pifithrin-α treatment reversed repression of *Bcl2* and induction of *Bak1* in *Neil*-deficient EBs + RA, confirming a TP53-dependent regulation of both genes (*Figure 12—figure supplement 1C*).

The apoptosis effector CASPASE-3 is induced upon- and required for mESC neural differentiation (*Fujita et al., 2008*). Concordantly, levels of CASPASE-3 (cleaved and uncleaved) were systematically decreased in *Neil*-deficient EBs + RA (*Figure 12—figure supplement 2A*), thus different from Caspase-3-effected apoptosis in *Xenopus* (*Figure 1I*). Levels of CASPASE-7, the alternative effector caspase of the intrinsic apoptosis pathway (*Lakhani et al., 2006*), were unchanged (*Figure 12—figure supplement 2B*).

Among the upregulated TP53-target genes in *Neil*-deficient EBs + RA were effectors of cell cycle arrest (e.g. *Cdkn1a*). We therefore tested for cell cycle differences in *Neil1-* and *Neil2*-deficient cells by flow cytometry analysis. However, while cell cycle profiles of EBs and EBs + RA were clearly distinguishable from mESCs (more G1-phase and fewer S- and G2/M-phase cells upon mESC differentiation; *White and Dalton, 2005*), there were no significant differences in cell cycle profiles between the control and *Neil1-* or *Neil2*-deficient cells, arguing against a TP53-induced cell cycle arrest (*Figure 12—figure supplement 2C*). In line, cell cycle effects are also absent after forced TP53 induction in neural crest cells in mice (*Bowen et al., 2019*).

Collectively, the results support a model in which NEIL1 and NEIL2 function as mitochondrial DNA repair glycosylases to counteract an increased oxidative stress during neurogenesis. Thereby, NEIL1 and NEIL2 protect against a mitochondrial-induced TP53-DDR and an intrinsic apoptosis pathway, and safeguard neural differentiation (*Figure 12E*).

## Discussion

One-third of all congenital birth defects are craniofacial malformations that arise by perturbations in cNCC development (*Sakai and Trainor, 2016*). Hence, understanding the environmental and genetic causes leading to perturbations of cNCC development is important for the development of potential therapeutic avenues for their prevention. The main finding of our study is the elucidation of a mechanism whereby disruption of the ubiquitous DNA glycosylases NEIL1 and NEIL2 leads to neural and cNCC differentiation defects during embryonic development. Our study indicates that cNCC defects caused by NEIL1- and NEIL2-deficiency are attributable primarily to their role in protecting against oxidative DNA lesions, in particular of the mitochondrial genome, rather than in promoting epigenetic DNA demethylation. Our study, therefore, links mitochondrial BER to neural cell differentiation.

The physiological role of NEIL1 and NEIL2 DNA glycosylases has previously been analyzed in mouse mutants. *Neil1* mutants present metabolic syndrome and show impaired brain function and neuronal stress resistance in adults (*Canugovi et al., 2012*; *Canugovi et al., 2015*; *Vartanian et al., 2006*). *Neil2*-null mice are susceptible to innate inflammation (*Chakraborty et al., 2015*). These abnormalities were accompanied by BER defects in both types of mutants. *Neil1,2,3* triple-mutant

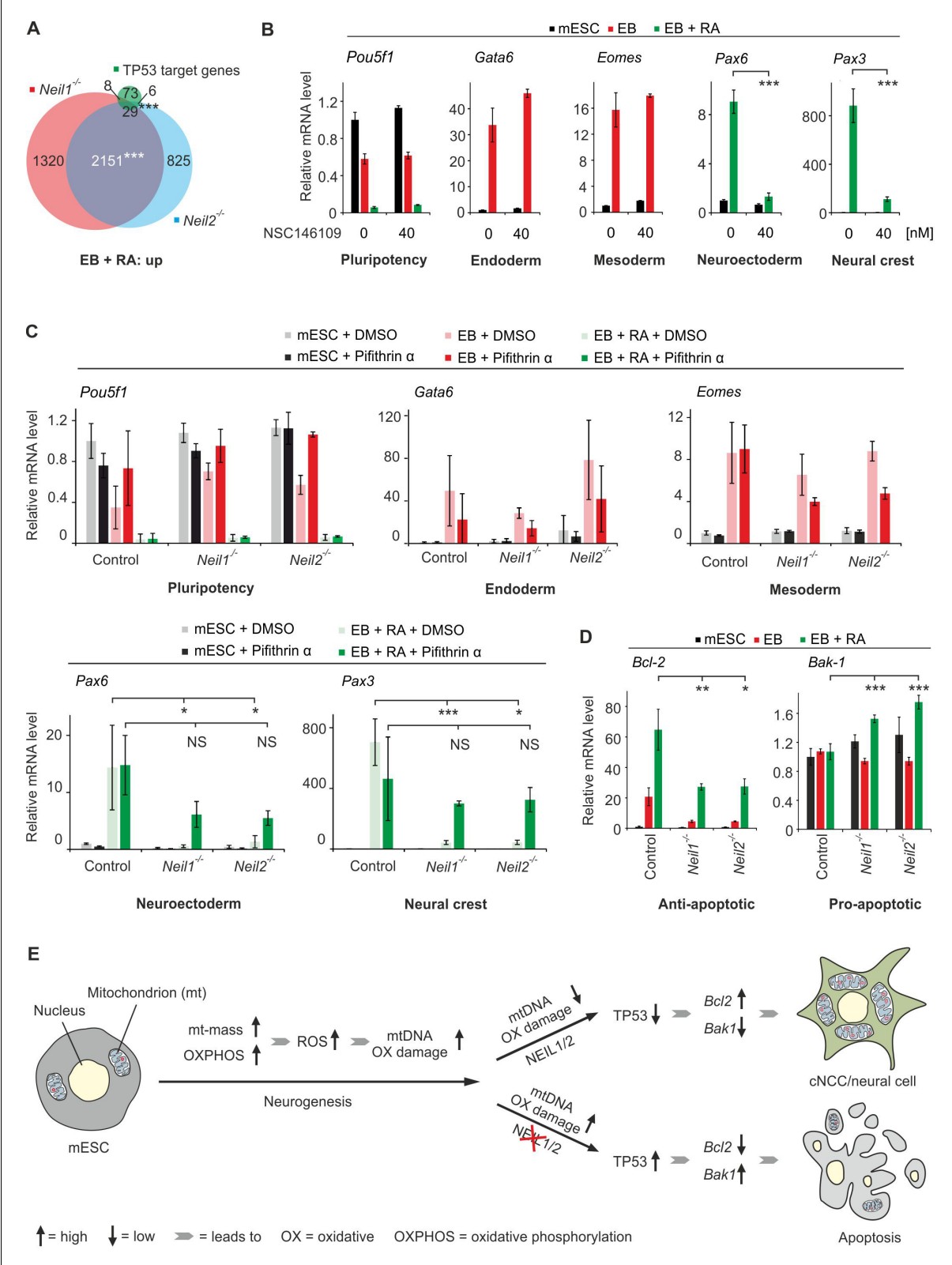

**Figure 12.** *Neil*-deficiency induces a TP53-mediated intrinsic apoptosis in embryoid bodies. (A) Overlap of upregulated genes from *Neil1* and *Neil2* single-deficient EBs + RA, and 116 direct TP53 target genes (**Fischer, 2017**). (B) qPCR expression analysis of the indicated marker genes in mock (DMSO) and NSC 146109- (TP53 stabilizer) treated control mESCs, EBs and EBs + RA. Expression of marker genes was normalized to *Tbp* and is relative to mock-treated mESCs (s.d., n = 3 technical replicates). (C) qPCR expression analysis of marker genes as in (B) in control, *Neil1* and *Neil2* single-mutant
*Figure 12 continued on next page*

*Figure 12 continued*

mESCs, EBs and EBs + RA treated with 50 μM Pifithrin-α or mock treated (DMSO). Expression of marker genes was normalized to *Tbp* and is relative to mock-treated control mESCs (s.d., n = 3 biological replicates). (**D**) qPCR expression analysis of *Bcl-2* and *Bak-1* in *Neil1* and *Neil2* single-deficient mESCs, EBs and EBs + RA. Expression of both genes was normalized to *Tbp* and is relative to control mESC clones (s.d., n = 3 biological replicates). (**E**) Model for the role of NEIL1 and NEIL2 in mESC differentiation towards cNCC/neural cells. Neurogenesis is accompanied by a metabolic switch from glycolysis to oxidative phosphorylation, a concomitant increase in mass and number of mitochondria per cell and hence escalated oxidative stress. High ROS levels render neural and neural crest cells particularly vulnerable to oxidative DNA damage and thus dependent on efficient damage repair, including by NEIL1 and NEIL2. Upon NEIL-deficiency, oxidative mtDNA damage induces apoptosis impairing neural and neural crest differentiation. ROS, reactive oxygen species.

DOI: https://doi.org/10.7554/eLife.49044.024

The following figure supplements are available for figure 12:

**Figure supplement 1.** TP53 response in *Neil*-deficient cells.

DOI: https://doi.org/10.7554/eLife.49044.025

**Figure supplement 2.** CASPASE levels and cell cycle profiles of *Neil*-deficient cells.

DOI: https://doi.org/10.7554/eLife.49044.026

---

mice were recently reported to be viable, but mice were only analyzed- and reported negative for cancer predisposition (*Rolseth et al., 2017*).

In contrast, we found that cNCC development showed a surprisingly specific vulnerability towards NEIL-deficiency both in *Xenopus* embryos (Neil2) as well as in differentiating mESCs (NEIL1 and NEIL2). Deficiency of NEIL3, which processes a different spectrum of lesions compared to NEIL1 and NEIL2 and is not found in mitochondria (*Prakash and Doublié, 2015*), had no effect on neural and cNCC development. The cNCC abnormalities in *Xenopus* were mirrored in the transcriptome of differentiating *Neil1*- and *Neil2*-deficient mESCs. In *Xenopus* embryos and mESCs, we elucidated the mechanism as a TP53-mediated DNA damage response, which induced apoptosis in *Xenopus* (Caspase-3-dependent) and mouse EBs (downregulation of *Bcl2* and induction of *Bak1*) without major effects on the cell cycle in both model systems. Moreover, in *Xenopus* and mESCs we found that elevated ROS reduced cNCC specification, supporting a conserved mechanism whereby an oxidative DNA damage response impairs cNCC differentiation.

Similarly, *Apex1* mutant mESCs and Apex1 *Xenopus* morphants displayed cNCC differentiation defects, although gene misregulation (mESCs) and malformations (*Xenopus*) were more severe than in *Neil*-mutant mESCs and Neil2 morphants, consistent with early embryonic lethality of *Apex1* mutant mice (*Ludwig et al., 1998*; *Xanthoudakis et al., 1996*). In contrast, neither did *Tdg*-deficiency affect cNCC differentiation in mESCs, nor did combined *Tdg/Neil1*-deficiency rescue cNCC differentiation defects, arguing against oxidative DNA demethylation as the primary cause of the differentiation defect.

The question arises, why are neural and neural crest cells particularly sensitive to oxidative DNA damage? Similarly, why is the systemic oxidative stress response to pyocyanin in *Xenopus* embryos limited to neuroectoderm (*Figure 4B*)? Neurogenesis is accompanied by a metabolic switch from glycolysis to oxidative phosphorylation and, hence, escalated oxidative stress (*Khacho and Slack, 2018*). High intrinsic ROS levels may render neural crest cells particularly vulnerable to oxidative DNA damage and thus dependent on efficient damage repair. Consistently, neuroectoderm expresses high levels of DNA repair factors (*Albino et al., 2011*) and of *Tp53* itself (*Cheng et al., 1997*; *Hoever et al., 1994*; *Rinon et al., 2011*; this study), suggesting a specific adaptation to a lesion-prone environment.

Intriguingly, we identified mitochondrial DNA as the primary target for oxidative DNA damage in the absence of NEIL1 and NEIL2 and specifically upon mESC neural differentiation. This suggests a model whereby NEIL1 and NEIL2 function to repair and protect the mitochondrial genome against oxidative damages that accompany the metabolic switch upon neural differentiation. Thereby, NEIL1 and NEIL2 shelter neural specification from an intrinsic, mitochondrial-induced apoptosis pathway (*Figure 12E*). NEIL1 and NEIL2 both localize in mitochondria besides the nucleus (*Hu et al., 2005*; *Mandal et al., 2012*). Moreover, *Neil1*-mutant mice harbor increased mtDNA damage in liver tissue, which might be related to the metabolic syndrome observed in these mice (*Vartanian et al., 2006*). Our study therefore corroborates the physiological relevance of NEIL1 and NEIL2 in mtDNA repair and ties mitochondrial BER to cell differentiation. Similarly, the observed neural and cNCC

differentiation defects of *Apex1*-mutant cells (*Figure 8*) and *Xenopus* Apex1 morphants (*Figure 5*) might partly be due to inefficient abasic site repair of mtDNA, as a truncated APEX1 is present in mitochondria (*Chattopadhyay, 2006*).

Our findings align with studies, which documented a role of TP53 in vertebrate neural crest formation. In chick embryos, TP53 stabilization decreases cNCC differentiation, while dominant-negative TP53 increases the number of cNCC progenitors (*Rinon et al., 2011*). TP53 activation in mouse embryos specifically causes severe neural crest defects (*Bowen et al., 2019*; *Van Nostrand et al., 2014*). Moreover, Treacher Collins syndrome (TCS), a congenital disorder characterized by severe cNCC and craniofacial anomalies, is caused by impaired ribosomal biogenesis due to deficiency in the Pol I transcription machinery, nucleolar dysfunction and/or rDNA damage, resulting in TP53 activation (*Calo et al., 2018*). Consequently, partial TP53 deficiency or pharmacologic TP53 inhibition ameliorates the craniofacial defects in TCS-mutants (*Jones et al., 2008*; *Sakai et al., 2016*), emphasizing the role of TP53 in cNCC development. Our results support the observation that neural tissue is particularly vulnerable to *Neil1*-deficiency in adults and is linked to neurodegenerative disease (*Canugovi et al., 2012*; *Canugovi et al., 2015*; *Vartanian et al., 2006*).

Why then do *Neil1*- and *Neil2*-mutant mice develop without obvious neural malformations? First, genetic compensation of *Neil*-mutants may occur via *Nthl1*, which has an overlapping substrate spectrum and also localizes both in the nucleus and mitochondria (*Ikeda et al., 2002*; *Jacobs and Schär, 2012*). Single-knockout mice deficient for *Neil1* and *Nthl1* are phenotypically inconspicuous, the combined knockout, however, is highly cancer prone, indicative of mutual compensation of both factors for oxidative DNA damage repair (*Chan et al., 2009*). mESC differentiation in vitro may not reproduce the complexity of in utero development where transcriptomic fine-tuning can buffer genetic ablations in the developing embryo. Second, differentiating mESCs in vitro experience gas phase oxygen partial pressure ($pO_2$) of 142 mmHg, whereas embryonic cells in vivo are exposed to $pO_2$ values of 0–30 mmHg (*Powers et al., 2008*). Similarly, one millimeter-sized *Xenopus* embryos developing close to the air-aquatic interface likely experience $pO_2$ levels closer to ambient partial pressure. Hence, cNCCs in differentiating mESCs as well as frog embryos have to cope with higher $pO_2$ and hence ROS levels than is the case for mouse embryos in utero. Increased basal ROS levels might sensitize cultured cells and frog embryos when challenged by *Neil*-deficiency during neural and cNCC differentiation. Third, apparently none of the *Neil*-mutant studies has specifically investigated whether *Neil*-deficient mice exhibit cNCC development-related cranial malformations, which may require special bone- and cartilage staining procedures for detection. Hence, our results call for a (re-)analysis of *Neil*-mutant mice for cranial abnormalities possibly under high fat diet, which favors basal ROS production (*Vial et al., 2011*).

Our findings also support the proposition that antioxidant supplementation may be beneficial for the prevention of craniofacial defects (*Sakai et al., 2016*), since environmental factors such as alcohol and nicotine promote ROS formation (*Wright et al., 1999*; *Zhao and Reece, 2005*). We also note that *Neil1* is one of 11 genes affected in a chromosomal micro-deletion of a patient presenting craniofacial defects (*Li and Bodamer, 2014*).

Finally, while our study demonstrates the importance of NEIL1 and NEIL2 to protect cNCCs against oxidative DNA damage, it does not exclude that NEIL1 and NEIL2 play a physiological role in TET/TDG-mediated gene regulation in other embryonic processes or adult tissues. Elucidating such direct gene-regulatory roles may require genome-wide monitoring of 5mC and its oxidation products in *Neil*-mutants. The here-established *Neil*-, *Apex1*- and *Tdg*-mutant mESCs will be useful for this and other investigations into the biology and mechanisms of BER enzymes.

## Materials and methods

**Key resources table**

| Reagent type (species) or resource | Designation | Source or reference | Identifiers | Additional information |
|---|---|---|---|---|
| Gene (*Homo sapiens*) | *NEIL1* | ORFeome clone collection | BC010876.1 | |

*Continued on next page*

*Continued*

| Reagent type (species) or resource | Designation | Source or reference | Identifiers | Additional information |
|---|---|---|---|---|
| Gene (*Homo sapiens*) | *NEIL2* | ORFeome clone collection | BC013964.2 | |
| Gene (*Homo sapiens*) | *APEX1* | ORFeome clone collection | BC008145.1 | |
| Gene (*Xenopus tropicalis*) | *bcl2l1* | Dharmacon | MXT1765-202788918 | |
| Strain, strain background (*Xenopus laevis*) | *Xenopus leavis* | Nasco | not available | |
| Cell line (*Mus musculus*) | E14TG2a | ATCC | CRL-1821 | murine embryonic stem cells |
| Cell line (*Mus musculus*) | E14TG2a clone Control #1 | this paper | | generated from E14TG2a |
| Cell line (*Mus musculus*) | E14TG2a clone Control #4 | this paper | | generated from E14TG2a |
| Cell line (*Mus musculus*) | E14TG2a clone Control #7 | this paper | | generated from E14TG2a |
| Cell line (*Mus musculus*) | E14TG2a clone Neil1,2,3-/- #23 | this paper | | generated from E14TG2a |
| Cell line (*Mus musculus*) | E14TG2a clone Neil1,2,3-/- #85 | this paper | | generated from E14TG2a |
| Cell line (*Mus musculus*) | E14TG2a clone Neil1,2,3-/- #93 | this paper | | generated from E14TG2a |
| Cell line (*Mus musculus*) | E14TG2a clone Neil1-/- #7 | this paper | | generated from E14TG2a |
| Cell line (*Mus musculus*) | E14TG2a clone Neil1-/- #9 | this paper | | generated from E14TG2a |
| Cell line (*Mus musculus*) | E14TG2a clone Neil1-/- #11 | this paper | | generated from E14TG2a |
| Cell line (*Mus musculus*) | E14TG2a clone Neil2-/- #1 | this paper | | generated from E14TG2a |
| Cell line (*Mus musculus*) | E14TG2a clone Neil2-/- #11 | this paper | | generated from E14TG2a |
| Cell line (*Mus musculus*) | E14TG2a clone Neil2-/- #14 | this paper | | generated from E14TG2a |
| Cell line (*Mus musculus*) | E14TG2a clone Neil3-/- #3 | this paper | | generated from E14TG2a |
| Cell line (*Mus musculus*) | E14TG2a clone Neil3-/- #23 | this paper | | generated from E14TG2a |
| Cell line (*Mus musculus*) | E14TG2a clone Neil3-/- #28 | this paper | | generated from E14TG2a |
| Cell line (*Mus musculus*) | E14TG2a clone Apex1-/- #46 | this paper | | generated from E14TG2a |
| Cell line (*Mus musculus*) | E14TG2a clone Tdg-/- #25 | this paper | | generated from E14TG2a |
| Cell line (*Mus musculus*) | E14TG2a clone Control #4 + Control #1 | this paper | | generated from E14TG2a clone Control #4 |
| Cell line (*Mus musculus*) | E14TG2a clone Neil1-/- #7 + Control #1 | this paper | | generated from E14TG2a clone Neil1-/- #7 |
| Cell line (*Mus musculus*) | E14TG2a clone Control #4 + Tdg-/- #7 | this paper | | generated from E14TG2a clone Control #4 |

*Continued on next page*

*Continued*

| Reagent type (species) or resource | Designation | Source or reference | Identifiers | Additional information |
|---|---|---|---|---|
| Cell line (*Mus musculus*) | E14TG2a clone Neil1-/- #7 + Tdg-/- #11 | this paper | | generated from E14TG2a clone Neil1-/- #7 |
| Cell line (*Mus musculus*) | E14TG2a clone Control #4 + pcDNA3.1_empty | this paper | | generated from E14TG2a clone Control #4 |
| Cell line (*Mus musculus*) | E14TG2a clone Neil1-/- #7 + pcDNA3.1_empty | this paper | | generated from E14TG2a clone Neil1-/- #7 |
| Cell line (*Mus musculus*) | E14TG2a clone Neil1-/- #7 + pcDNA3.1_NEIL1-2x FLAG (active) | this paper | | generated from E14TG2a clone Neil1-/- #7 |
| Cell line (*Mus musculus*) | E14TG2a clone Neil1-/- #7 + pcDNA3.1_2xFLAG-NEIL1 (inactive) | this paper | | generated from E14TG2a clone Neil1-/- #7 |
| Cell line (*Mus musculus*) | E14TG2a clone Neil2-/- #11 + pcDNA3.1_empty | this paper | | generated from E14TG2a clone Neil2-/- #11 |
| Cell line (*Mus musculus*) | E14TG2a clone Neil2-/- #11 + pcDNA3.1_NEIL2-2xFLAG (active) | this paper | | generated from E14TG2a clone Neil2-/- #11 |
| Cell line (*Mus musculus*) | E14TG2a clone Neil2-/- #11 + pcDNA3.1_2xFLAG-NEIL2 (inactive) | this paper | | generated from E14TG2a clone Neil2-/- #11 |
| Antibody | mouse monoclonal anti-alpha tubulin | Sigma | T5168 | (1: 1000) |
| Antibody | rabbit polyclonal anti-NEIL1 | Abcam | ab21337 (discontinued) | (1: 500) |
| Antibody | rabbit polyclonal anti-NEIL2 | Abcam | ab124106 (discontinued) | (1: 1000) |
| Antibody | rabbit polyclonal anti-APE1 | Abcam | ab137708 | (1: 1000) |
| Antibody | rabbit polyclonal anti-phospho-Chk1 (Ser345) | Cell Signaling | #2341 | (1: 1000) |
| Antibody | mouse monoclonal anti-p53 (X77) | Thermo Fisher Scientific | MA1-12549 | (1: 1000) |
| Antibody | rabbit polyclonal anti-histone H3 | Abcam | ab1791 | (1: 5000) |
| Antibody | rabbit polyclonal anti-histone H3 (Ser10) | Sigma | 06–570 | (1: 500) |
| Antibody | rabbit polyclonal anti-TDG | Active Motif | 61437 | (1: 1000) |
| Antibody | rabbit polyclonal anti-Caspase-3 | Cell Signaling | #9662 | (1: 1000) |
| Antibody | rabbit polyclonal anti-Caspase-7 | Cell Signaling | #9492 | (1: 1000) |
| Antibody | HRP-coupled goat polylonal anti rabbit IgG | Dianova | 111-035-144 | (1: 10000) |

*Continued on next page*

*Continued*

| Reagent type (species) or resource | Designation | Source or reference | Identifiers | Additional information |
|---|---|---|---|---|
| Antibody | HRP-coupled goat polyclonal anti mouse IgG | Dianova | 115-035-146 | (1: 10000) |
| Recombinant DNA reagent | pCS2FLAG | Addgene | RRID:Addgene_16331 | |
| Recombinant DNA reagent | pcDNA3.1(+) | Invitrogen | V79020 | |
| Recombinant DNA reagent | pX330-U6-Chimeric_BB-CBh-hSpCas9 | Addgene | RRID:Addgene_42230 | |
| Recombinant DNA reagent | pPGKPuro | Addgene | RRID:Addgene_11349 | |
| Recombinant DNA reagent | pCS105-xp53 | Stefano Piccolo | NA | |
| Peptide, recombinant protein | APE 1 | NEB | M0282 | |
| Peptide, recombinant protein | Endonuclease III (Nth) | NEB | M0268 | |
| Commercial assay or kit | RNeasy Mini Kit | Qiagen | 74104 | |
| Commercial assay or kit | DNeasy Blood and Tissue Kit | Qiagen | 69504 | |
| Commercial assay or kit | Blood and Cell Culture DNA Midi Kit | Qiagen | 13343 | |
| Commercial assay or kit | RNA 6000 Nano kit | Agilent | 5067–1511 | |
| Commercial assay or kit | Qubit dsDNA HS Assay Kit | Invitrogen | Q32851 | |
| Commercial assay or kit | TruSeq RNA Sample Preparation v2 Kit | Illumna | RS-122–2001/ RS-122–2002 | |
| Commercial assay or kit | MEGAscript SP6 Transcription kit | Invitrogen | AM1330 | |
| Chemical compound, drug | Pyocyanin | Sigma | P0046 | |
| Chemical compound, drug | L-Ascorbic acid 2-phosphate sesquimagnesium salt hydrat | Sigma | A8960 | |
| Chemical compound, drug | Pifithrin α | Sigma | P4359 | |
| Chemical compound, drug | NSC 146109 | Santa Cruz Biotechnology | sc-203652 | |
| Chemical compound, drug | 2,6-Di-*tert*-butyl-4-methylphenol (BHT) | Sigma | B1378 | |
| Chemical compound, drug | deferoxamine mesylate | Sigma | D9533 | |
| Software, algorithm | LightCycler 480 software | Roche | 4994884001 | |
| Software, algorithm | bcl2fastq Conversion Software v.1.8.4 | Illumina | http://emea.support.illumina.com/downloads/bcl2fastq_conversion_software_184.html | |

*Continued on next page*

*Continued*

| Reagent type (species) or resource | Designation | Source or reference | Identifiers | Additional information |
|---|---|---|---|---|
| Software, algorithm | FastQC | Babraham Bioinformatics | https://www.bioinformatics.babraham.ac.uk/projects/fastqc/ | |
| Software, algorithm | STAR v.2.5.4b | PMID: 23104886 | | |
| Software, algorithm | Subread feature Counts v.1.5.1 | PMID: 24227677 | | |
| Software, algorithm | DESeq2 | PMID: 25516281 | | |
| Software, algorithm | PANTHER | The Gene Ontology Resource | http://pantherdb.org | |
| Software, algorithm | TopHat v. 2.0.9 | Johns Hopkins University | https://ccb.jhu.edu/software/tophat/index.shtml | |
| Software, algorithm | iGenomes | Illumina | http://emea.support.illumina.com/sequencing/sequencing_software/igenome.html | |
| Software, algorithm | BioVenn | PMID: 18925949 | http://www.biovenn.nl/ | |
| Software, algorithm | WebGestalt | PMID: 28472511 | http://webgestalt.org/ | |
| Software, algorithm | HTSeq-count v. 0.5.4 | | https://htseq.readthedocs.io/en/release_0.11.1/ | |
| Software, algorithm | Xenbase | PMID: 29059324 | ftp://ftp.xenbase.org/pub/Genomics/JGI/Xenla9.2 | |
| Software, algorithm | MassHunter Quantitative Analysis, v. B.05.02 | Agilent Technologies | https://www.agilent.com/en/products/software-informatics/masshunter-suite/masshunter/masshunter-software | |
| Software, algorithm | FACSDiva | BD | http://www.bdbiosciences.com/en-us/instruments/research-instruments/research-software/flow-cytometry-acquisition/facsdiva-software | |
| Software, algorithm | FlowJo software v. 10.5.3 | BD | https://www.flowjo.com/solutions/flowjo/downloads/previous-versions | |

## Expression constructs

Human *NEIL1*, *NEIL2* and *APEX1* cDNAs (BC010876.1, BC013964.2 and BC008145.1, respectively) were from the ORFeome clone collection. For in vitro transcription *NEIL2* and *APEX1* cDNA was inserted into pCS2FLAG (Addgene plasmid 16331). Additionally, *NEIL1* and *NEIL2* cDNAs were inserted into pcDNA3.1(+) (Invitrogen) as C-terminal (catalytically active) and N-terminal (catalytically inactive) 2xFLAG-tag expression constructs. pCMV-Sport6-xt.bcl2l1 encoding wild type *Xenopus bcl2l1* was purchased from Dharmacon (MXT1765-202788918). pCS105-xp53 encoding wildtype *Xenopus tp53* was a kind gift from S. Piccolo (University of Padua, Italy).

## Immunoblotting

Western blot analysis of *X. laevis* samples was essentially as described (*Kirsch et al., 2017*). Mouse embryoid bodies were incubated in lysate buffer (20 mM Tris pH 7.5, 150 mM NaCl, 5 mM EDTA,

2% NP-40 and Complete Mini Protease Inhibitor Cocktail (Roche)). Lysates were cleared by centrifugation and protein concentrations were estimated by bicinchoninic acid (BCA) assay using BSA as standard followed by SDS-PAGE and western blotting. Antibodies are depicted in key resources table.

## Reverse-transcriptase coupled quantitative real time PCR (RT-qPCR)

Total RNA was prepared by RNeasy Mini Kit (Qiagen) including an on-column DNase digestion according to the manufacturer's instructions. Complementary DNA (cDNA) was synthesized using SuperScript II reverse transcriptase (Life Technologies). Quantitative real time PCR was performed on a LightCycler 480 (Roche) in technical duplicates using the Universal ProbeLibrary technology (Roche) including the supplier's LightCycler 480 Probes Master. Quantitative analysis was performed with LightCycler 480 software (Roche). Primer sequences and hydrolysis probe numbers are listed in *Supplementary file 6*.

## *X. laevis* embryo manipulation and staining

Animal experiments with *X. laevis* were approved by state authorities (Landesuntersuchungsamt Rheinland-Pfalz, reference number 23177–07/A12-5-001). No blinding or randomization was performed. Embryos were obtained by in vitro fertilization as described (*Gawantka et al., 1995*) and cultivated in 0.1x Barth's solution (*Wang et al., 2010*). Human and *Xenopus* expression constructs as depicted above were used as templates to generate mRNAs with the MEGAscript SP6 Transcription Kit (Invitrogen) according to the manufacturer's instructions. Morpholino antisense oligonucleotides (MOs, see *Supplementary file 6*) were designed to block translation of the respective gene. MOs and mRNAs were injected two times into animal blastomeres at one-cell stage with a total volume of 10 nl per embryo. For overexpression, each embryo was injected with: human *NEIL2* mRNA, 2 ng; *Xenopus tp53* mRNA, 200 pg; *Xenopus bcl2l1* mRNA, 2 ng. Total amounts of single MOs injected per embryo were as follows: *neil2* MO, 40 ng and *tp53* MO, 20 ng. For double MO injections, each embryo was injected with a mixture of 40 ng *neil2* MO and 20 ng *p53* MO. For cartilage staining, one blastomere of two-cell stage embryos was injected with 5 nl (total volume) of *neil2* MO (7.5 ng) and human *NEIL2* mRNA (375 pg). Control mRNA used for injections was *preprolactin*. For ROS induction embryos were grown in 0.1x Barth's solution supplemented with pyocyanin (Sigma, P0046; final concentration 10–25 µM). For Vitamin C treatment embryos were grown in 0.1x Barth's solution supplemented with 100 µM L-Ascorbic acid 2-phosphate (Sigma, A8960). Embryos were fixed at the indicated developmental stage in freshly prepared MEMFA (100 mM MOPS pH 7.4, 2 mM EGTA, 1 mM $MgSO_4$, 4% formaldehyde) for 1 hr at RT. After fixation, embryos were washed twice in 100% ethanol at RT for 5 min and stored in 100% ethanol at −20℃. Whole mount in situ hybridization was performed as described (*Bradley et al., 1996*). In situ hybridization probes were generated from cDNAs for *X. laevis* neuronal marker genes, *ccng* and *tp53* using the Dig RNA labeling Kit (Roche). For lineage tracing, *lacZ* mRNA was co-injected (250 pg/blastomere) and β-gal staining was performed as described (*Sive et al., 2000*) using X-gal as substrate. Neural plates were dissected at stage 14 with Dumont No. five forceps. Cartilage staining was performed as described (*Nie and Bronner, 2015*). TUNEL assays were carried out as previously described (*Hensey and Gautier, 1998*). Images were taken on a Zeiss SteREO Discovery.V20 microscope.

## Cell culture

Mouse E14TG2a embryonic stem cells (mESCs) were obtained from ATCC, number CRL-1821. Identity has been authenticated by ATCC. E14TG2a were initially tested Mycoplasma-positive, decontaminated using MycoZap Elimination Reagent (Lonza) and subsequently used for the study. E14TG2a cells were cultured on plates coated with 0.1% Gelatin (Millipore) in DMEM supplemented with 15% PANSera ES FBS (PAN Biotech), 2 mM L-Glutamine, 100 µM non-essential amino acids (NEAA, Gibco), 1 mM sodium pyruvate (Gibco), 100 µM 2-mercaptoethanol (Sigma), 1000 U/ml Leukemia inhibitory factor (LIF, Millipore), 100 U/ml PEN-STREP at 37℃ in 5% $CO_2$ and 20% $O_2$.

## CRISPR/Cas9-mediated gene deletions

To generate mESCs deficient for *Neil1,2,3*, *Neil1*, *Neil2*, *Neil3*, *Apex1*, *Tdg*, and *Neil1/Tdg*, $1 \times 10^6$ mESCs were seeded and transfected the next day with either empty or gRNA encoding pX330-U6-

Chimeric_BB-CBh-hSpCas9 (Addgene #42230) mixed with the selection plasmid pPGKPuro (Addgene #11349) using Lipofectamine 2000 (Invitrogen) according to manufacturer's instruction. Cells were selected with 2 µg/ml puromycin for 6 days, colonies picked, passaged and subjected to genotyping PCR using primers flanking the anticipated deletion region (see *Supplementary file 6*). Positive clones were expanded for further analyses.

## Embryoid body (EB) differentiation

$3.5 \times 10^6$ mESCs were plated on non-adherent 10 cm bacterial dishes (Greiner) in 15 ml CA medium (*Bibel et al., 2007*). CA medium was changed every second day. For retinoic acid induced neural EB differentiation, CA medium was supplemented with all-*trans*-Retinoic acid (R2625, Sigma; final concentration 5 µM) at days 4 and 6 of differentiation. EBs were harvested after 8 days of differentiation. For EB differentiation in presence of ROS inducer pyocyanin (Sigma, P0046; final concentration 2 µM), TP53 stabilizer NSC 146109 (Santa Cruz Biotechnology, sc-203652; final concentration 40 nM) and TP53 inhibitor Pifithrin-α (Sigma, P4359; final concentration 50 µM) drug treatment of cells was started 24 hr prior to plating mESCs on non-adherent dishes and was continued throughout differentiation.

Stable transfection mESCs were transfected with empty vector or human *NEIL1* and *NEIL2* expressing pcDNA3.1 constructs (see 'Expression constructs') using Lipofectamine 2000 (Invitrogen) according to manufacturer's instructions. Following selection for 6 days with 500 µg/ml G-418 (Gibco) single colonies were picked, expanded in selection medium and analyzed by RT qPCR for expression of the transgene.

## Teratoma assay

Transplantation of mESCs into immunodeficient (NSG) mice, animal husbandry and tumor preparation was performed by EPO Berlin GmbH. Resulting tumors were split and either shock frozen, or formalin fixed, paraffin embedded, sectioned and stained with hematoxylin and eosin for histological analysis. Tumors from each three independent control and $Neil1,2,3^{-/-}$ mESC lines were grown in technical triplicates.

## RNA sequencing
### Frog embryos
Control MO and *neil2* MO-injected *Xenopus laevis* embryos in triplicates ($n \geq 5$ per batch of embryo) at stage 23 were lysed with 700 µl Qiazol reagent and homogenized by pipetting. After 5 min incubation at room temperature, 200 µl chloroform was added and samples were shaken vigorously. RNA was isolated subsequently using a Qiagen RNeasy Mini Kit according to the manufacturer's instructions. RNA integrity was validated using an RNA 6000 Nano kit on an Agilent 2100 Bioanalyzer. NGS library preparation was performed using Illumina's TruSeq RNA Sample Preparation v2 Kit followed the standard protocol. Libraries were prepared with a starting amount of 1 µg, amplified in 12 PCR cycles, profiled in a DNA 1000 chip on an Agilent 2100 Bioanalyzer and quantified using the Qubit dsDNA HS Assay Kit in a Qubit 2.0 Fluorometer (Life Technologies). All six libraries were pooled in equimolar ratio and sequenced on HiSeq 2000 in single read mode for 50 cycles plus additional eight cycles for the index read. Sample demultiplexing and FastQ file generation was performed using Illumina's bcl2fastq Conversion Software v.1.8.4. The raw sequence reads were quality assessed using FastQC (https://www.bioinformatics.babraham.ac.uk/projects/fastqc/) and aligned to the *Xenopus laevis* v.9.2 genome assembly with JGI gene annotation from Xenbase (ftp://ftp.xenbase.org/pub/Genomics/JGI/Xenla9.2) using STAR v.2.5.4b (*Dobin et al., 2013*) with option '–outFilterMismatchNmax 2'. The mapped reads were summarised on the gene level using Subread featureCounts v.1.5.1 (*Liao et al., 2014*) with default parameters. Differential gene expression analysis was performed with the Bioconductor package DESeq2 v.1.18.1 (*Love et al., 2014*) following the recommended analysis workflow with independent gene filtering. Differentially expressed genes were identified using a statistical cutoff of 10% false discovery rate (FDR) and an effect size filter 'log2 fold change (FC)' above 0.5. Differentially expressed up- and downregulated genes at log2FC >1 were subjected to pathway enrichment analysis with PANTHER (http://pantherdb.org) using the *Xenopus laevis* gene symbols without L/S alloallele suffixes, background list of all DESeq2-tested genes, human pathway annotation and the default enrichment cutoff of 5% FDR.

## Mouse cells

Snap frozen teratomas derived from each three control and *Neil1,2,3*$^{-/-}$ mESC lines in triplicates were homogenized with an Ultra-TURRAX disperser (IKA) in presence of 2 ml TRIzol reagent (Invitrogen) using ~200 mg tissue per tumor. After clearance of supernatant by centrifugation for 5 min at 12.000 x g at 4°C, 0.4 ml of chloroform was added and the solution centrifuged for 15 min at 12.000 x g at 4°C. RNA from the aqueous phase was precipitated with 1 ml isopropanol, washed with 75% ethanol, air dried and resuspended in RNase-free water. Contaminating DNA was digested with DNase I followed by an RNA cleanup (RNeasy kit, Qiagen) according to manufacturer's instructions. Total RNA from control, *Neil1*$^{-/-}$, *Neil2*$^{-/-}$ mESCs, EBs and EBs+RA in biological triplicates and *Apex1*$^{-/-}$ mESCs, EBs and EBs + RA in technical duplicates was isolated using the Qiagen RNeasy Mini Kit according to the manufacturer's instructions, including on-column DNase I digest.

RNA was quantified with a Thermo NanoDrop and quality tested on Agilent 2100 Bioanalyzer. Only samples with RIN values > 9 were used for RNA-seq. NGS libraries were prepared from total RNA using the TruSeq RNA Sample Prep Kit v2 (Illumina) according to the manufacturer's recommendations and amplified in 12 PCR cycles. Libraries were profiled in a High Sensitivity DNA chip on a 2100 Bioanalyzer and quantified using the Qubit dsDNA HS Assay Kit, in a Qubit 2.0 Fluorometer (Life Technologies). The NGS libraries were sequenced on a HiSeq 2000 Illumina sequencer, for 51 cycles plus seven cycles for the index read. Raw reads were quality assessed using FastQC (https://www.bioinformatics.babraham.ac.uk/projects/ fastqc/) and mapped to the mouse genome assembly NCBIM37/mm9 using TopHat v. 2.0.9 (https://ccb.jhu.edu/software/tophat) and a GTF gene annotation file from Illumina iGenomes. HTSeq-count v. 0.5.4 (https://htseq.readthedocs.io/en/release_0.11.1/) was used for summarizing the mapped reads on genes. The Bioconductor package DESeq2 (https://bioconductor.org/packages/release/bioc/html/DESeq2.html) was used to identify genes with significant differential expression in *Neil1,2,3*$^{-/-}$ teratomas, and *Neil1*, *Neil2* and *Apex1* single-knockout samples compared to the control samples with a statistical cutoff of 10% false discovery rate (FDR) and an effect size filter log2FC > 0.5. Overlap of differentially expressed genes was created with BioVenn (*Hulsen et al., 2008*). Pathway enrichment analysis was performed with WebGestalt (*Wang et al., 2017*) using 'Wikipathway' as functional database with the default 5% FDR enrichment cutoff, differentially expressed up- and downregulated genes at log2FC > 1 and background list of all DESeq2-tested genes.

The RNA-seq datasets have been deposited in the NCBI GEO database under accession number GSE130082.

## Preparation of genomic, mitochondrial and synthetic DNA for LC-MS/MS analysis

Genomic DNA for analysis of 5mC, 5hmC, 5fC and 5caC was prepared using the DNeasy Kit (Qiagen) according to manufacturer's instructions.

Genomic DNA for abasic site analysis was prepared with the Qiagen Blood and Cell Culture DNA Midi Kit essentially as described (*Rahimoff et al., 2017*) but using buffers G2, QC, QF supplemented with each 400 µM of the antioxidants 2,6-Di-*tert*-butyl-4-methylphenol (BHT, Sigma B1378) and deferoxamine mesylate (DFOM, Sigma D9533). DNA was stored at −20°C in $H_2O$ supplemented with 40 µM BHT and DFOM.

Preparation of mtDNA was performed using the QIAprep Spin Miniprep Kit (Qiagen) following manufacturer's instructions. Buffers P1, P2, N3, PB and PE were supplemented with 400 µM BHT and DFOM. DNA was eluted with $H_2O$ containing 40 µM BHT and DFOM. Quantitative real time PCR as described above was used to calculate the enrichment of mtDNA over gDNA by the ΔΔCp method (*Quispe-Tintaya et al., 2013*). PCR mixture contained 1 ng of DNA and gDNA- and mtDNA-specific primers (mmActB and mmCytB, see *Supplementary file 6* for sequences).

Endogenous abasic sites on genomic and mtDNA (3 µg each) were processed by incubation with 20 units APE 1 (NEB, M0282) in a 50 µl reaction volume for 2 hr at 37°C. After phenol/chloroform extraction, DNA was further incubated with 20 units Endonuclease III (Nth, NEB, M0268) in a 50 µl reaction volume for 2 hr at 37°C, phenol/chloroform extracted, ethanol precipitated and resuspended in $H_2O$ supplemented with 40 µM BHT and DFOM.

Synthetic oligonucleotides were resuspended in $H_2O$ supplemented with 40 µM BHT and DFOM. The unmodified (40mer) and 5hU-containing oligo (40mer_5hU) were hybridized in 1x SSC (150 mM

NaCl, 15 mM trisodium citrate, pH 7.0) to the complementary strand (40mer_complementary) in a 1:1 molar ratio. For abasic site production 500 pmoles of the single-stranded uracil-containing oligo (40mer_U) were incubated with 25 units UDG (NEB, M0280) in a 50 µl reaction volume for 30 min at 37˚C. After phenol/chloroform extraction and ethanol precipitation the abasic site oligo was hybridized to the complementary strand as described above. APE 1- and Endonuclease III-treatment was performed as depicted above with 3 µg of double-stranded oligonucleotides. DNA was purified by phenol/chloroform, ethanol precipitated and resuspended in $H_2O$ supplemented with 40 µM BHT and DFOM for abasic site derivatization as outlined below. Oligonucleotide sequences are listed in *Supplementary file 6*.

### Quantitative mass spectrometry (LC-MS/MS)

Quantification of 5mC and oxidative derivatives was carried out as described before (*Schomacher et al., 2016*).

Quantification of abasic sites by LC-MS/MS was performed according to the published protocol (*Rahimoff et al., 2017*) with specific changes: Derivatization was performed with 1–2 µg of DNA and 30 nmoles of reagent 1a for 60 min at 37˚C. After an additional incubation with 30 nmoles of reagent 1a for 60 min at 37˚C reaction was stopped, the DNA ethanol precipitated, dissolved in 15 µl $H_2O$ and digested as described (*Schomacher et al., 2016*). After digest, DNA was mixed with an equal volume of isotopic standards, and 5 µL were injected for LC-MS/MS analysis. The chromatographic separation was performed on a ZORBAX SB-C18 column (Agilent, 5 µm, 2.1 ×50 mm). Elution was performed with 5 mM Ammonium acetate pH 6.9 and Acetonitrile (ACN), the flow rates were 0.4 ml/min for 0–7 min, 0.5 ml/min for 7–9 min and 0.4 ml/min for 9–10 min at 30˚C with the following gradient: 0–2 min, 0% ACN; 2–5 min, 0–5% ACN; 5–9 min, 5–50% ACN; 9–10 min, 0% ACN. Transitions corresponding to dG, 9a_1, 10a_1 and their respective isotopic standards $^{15}N_5$-dG, 9b_1, 10b_1 were monitored (for compounds 9a_1, 9_b1, 10_a1, 10_b1 see *Rahimoff et al., 2017*). The source-dependent parameters were as follow: gas temperature 110˚C, gas flow 19 l/min ($N_2$), Nebulizer 25 psi, sheath gas heater 375˚C, sheath gas flow 11 l/min ($N_2$), capillary voltage 2000 V (positive mode), nozzle voltage 0 V, fragmentor voltage 300 V. Compound dependent parameters were as previously described (*Rahimoff et al., 2017*) except that MS1 resolution for dG and $^{15}N_5$-dG were enhanced and MS2 was unit. For the rest of ions all MS1 and MS2 resolution were set to unit. Abasic sites were initially quantified over total dG and subsequently calculated over total N using a GC content of 42% of the mouse genome.

### Detection of mtDNA damage by quantitative PCR

Preparation of mtDNA was performed as described above. Quantitative PCR was essentially as described (*Gureev et al., 2017*) using 500 pg of mtDNA on a LightCycler 480 (Roche) in technical duplicates. Short fragments were amplified each with 20 s, long fragments with 2 min elongation steps in 35 cycles. SYBR Green was from Sigma (S9430). Calculation of lesions per 10 kb was as described (*Gureev et al., 2017*). For primer sequences and amplicon lengths see *Supplementary file 6*.

### Flow cytometry analysis

Cells were detached using 0.25% trypsin and washed with PBS containing 1% ESC grade FBS. Cells were fixed by adding dropwise ice-cold ethanol and subsequent incubation at −20˚C for 30 min or storage at this point. For propidium iodide staining, cells were washed twice with PBS containing 0.1% ESC grade FBS and 100 µg/ml RNAse A (Qiagen). Cells were then resuspended in PBS containing 50 µg/ml propidium iodide (Sigma) according to the cell number and incubated at room temperature for 10 min in the dark. Stained cells were then analysed by the BD LSRFortessaSORP flow cytometry system using FACSDiva software. Data analysis was performed with FlowJo software v. 10.5.3 (BD).

### Statistical analysis

Data presented are displayed as arithmetic mean, error bars represent standard deviations (s.d.) of the indicated replicates. Statistical significance as shown in bar diagrams was determined by two-tailed unpaired Student's t-test. Significance of overlapping groups of genes presented in Venn

Diagrams was calculated by hypergeometric distribution (http://nemates.org/MA/progs/overlap_stats.html) using a total number of 17 000 expressed genes per calculation. In triple overlaps significance was calculated on the basis of commonly deregulated genes in *Neil1* and *Neil2*-deficient cells. Significances are displayed as *p<0.05, **p<0.01, ***p<0.005. NS, not significant.

## Acknowledgements

We are grateful for technical support by the IMB Core Facilities Genomics, Bioinformatics, Microscopy and Flow Cytometry. We thank T Dehn (IMB) for animal care taking, R Rahimoff, E Korytiakova and T Carell (LMU Munich) for LC-MS/MS reagents, and S Piccolo (University of Padua) for reagents. This work was funded by the Deutsche Forschungsgemeinschaft (DFG, German Research Foundation) – project numbers NI286/17-1 and SFB 1361–03.

## Additional information

### Funding

| Funder | Grant reference number | Author |
| --- | --- | --- |
| Deutsche Forschungsgemeinschaft | NI286/17-1 | Christof Niehrs |
| Deutsche Forschungsgemeinschaft | 393547839 - SFB 1361, sub-project 03 | Christof Niehrs |

The funders had no role in study design, data collection and interpretation, or the decision to submit the work for publication.

### Author contributions

Dandan Han, Conceptualization, Data curation, Formal analysis, Validation, Investigation, Visualization, Methodology, Writing—original draft, Writing—review and editing; Lars Schomacher, Conceptualization, Data curation, Formal analysis, Supervision, Validation, Investigation, Visualization, Methodology, Writing—original draft, Writing—review and editing; Katrin M Schüle, Data curation, Formal analysis, Validation, Investigation, Visualization, Methodology, Writing—review and editing; Medhavi Mallick, Michael U Musheev, Emil Karaulanov, Data curation, Formal analysis, Validation, Investigation, Methodology, Writing—review and editing; Laura Krebs, Annika von Seggern, Data curation, Formal analysis, Investigation, Methodology; Christof Niehrs, Conceptualization, Supervision, Funding acquisition, Writing—original draft, Writing—review and editing

### Author ORCIDs

Dandan Han https://orcid.org/0000-0002-0585-2451
Lars Schomacher https://orcid.org/0000-0002-3841-5258
Katrin M Schüle https://orcid.org/0000-0001-6642-0030
Michael U Musheev https://orcid.org/0000-0002-0499-9689
Christof Niehrs https://orcid.org/0000-0002-9561-9302

### Ethics

Animal experimentation: Animal experiments with *X. laevis* were approved by state authorities (Landesuntersuchungsamt Rheinland-Pfalz, reference number 23177-07/A12-5-001).

### Decision letter and Author response

Decision letter https://doi.org/10.7554/eLife.49044.037
Author response https://doi.org/10.7554/eLife.49044.038

## Additional files

### Supplementary files

• Supplementary file 1. Differentially expressed genes in *Neil2* MO vs. control MO stage 23 *Xenopus laevis* embryos.
DOI: https://doi.org/10.7554/eLife.49044.027

• Supplementary file 2. Differentially expressed genes in *Neil1,2,3*-triple knockout compared to control teratomas.
DOI: https://doi.org/10.7554/eLife.49044.028

• Supplementary file 3. Differentially expressed genes in *Neil1*-knockout compared to control mESCs, EBs and EBs + RA.
DOI: https://doi.org/10.7554/eLife.49044.029

• Supplementary file 4. Differentially expressed genes in *Neil2*-knockout compared to control mESCs, EBs and EBs + RA.
DOI: https://doi.org/10.7554/eLife.49044.030

• Supplementary file 5. Differentially expressed genes in *Apex1*-knockout compared to control mESCs, EBs and EBs + RA.
DOI: https://doi.org/10.7554/eLife.49044.031

• Supplementary file 6. Oligonucleotides used in this study.
DOI: https://doi.org/10.7554/eLife.49044.032

• Transparent reporting form
DOI: https://doi.org/10.7554/eLife.49044.033

### Data availability

The RNA-seq datasets have been deposited in the NCBI GEO database under accession number GSE130082.

The following dataset was generated:

| Author(s) | Year | Dataset title | Dataset URL | Database and Identifier |
|---|---|---|---|---|
| Dandan Han, Lars Schomacher, Katrin M. Schüle, Medhavi Mallick, Michael U. Musheev, Emil Karaulanov, Laura Krebs, Annika von Seggern, and Christof Niehrs | 2019 | NEIL1 and NEIL2 DNA glycosylases protect against oxidative stress-induced inhibition of neural crest development | https://www.ncbi.nlm.nih.gov/geo/query/acc.cgi?acc=GSE130082 | NCBI Gene Expression Omnibus, GSE130082 |

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
