## [Decision Letter]

Thank you for submitting your article "NEIL1 and NEIL2 DNA glycosylases protect neural crest development against mitochondrial oxidative stress" for consideration by *eLife*. Your article has been reviewed by three peer reviewers, and the evaluation has been overseen by Marianne Bronner as the Senior and Reviewing Editor. The following individuals involved in review of your submission have agreed to reveal their identity: Eliezer Calo (Reviewer #1); Fadel Tissir (Reviewer #2).

The reviewers have discussed the reviews with one another and the Reviewing Editor has drafted this decision to help you prepare a revised submission.

Summary:

In this manuscript, Han et al. investigate the role of NEIL family bifunctional DNA glycosylases/lyases in *Xenopus* embryos and differentiating mouse embryonic stem cells (mESCs). They find that in *Xenopus* embryos Neil2 deficiency leads to defects in cranial neural crest cell development and link this to an oxidative stress-induced TP53-dependent mitochondrial DNA damage response that leads to apoptosis. They then use mESCs to confirm similar findings in mammalian cells. They suggest a model whereby NEIL1/NEIL2 function to repair and protect the mitochondrial genome against oxidative damages that accompany the metabolic switch upon neural differentiation. The manuscript presents an interesting connection between mitochondria oxidative stress and neural crest differentiation defects. As presented however, the study needs to be strengthened.

Essential revisions:

1) The authors must drop the claim that there is an effect on neural differentiation in *Xenopus* or do the work needed to validate this claim. e.g. look at multiple markers at both neural plate stages and later tailbud stages at neural differentiation markers.

2) The authors must use multiple markers in mESCs to validate effects on neural crest and neural differentiation.

3) The authors should examine *tp53* expression in NEIL2 morphants and pyocyanin treated embryos and for pyocyanin treated embryos drug treatment should be done more proximal to the time of neural crest/neural plate formation.

4) More rigorous validation of the CRISPR-Cas9 mutations is needed.

5) The authors need to provide additional evidence demonstrating that the oxidative stress response directly causes the phenotype and originates in the mitochondria. To me this is the most novel aspect of the work.

The full reviews are attached to provide further details, which we advise you to address if possible.

Reviewer #1:

This is an intriguing study that reveals how DNA glycosylases, particularly Neil1 and Neil2, protect neural crest cells from oxidative damage to mitochondrial DNA. The most novel aspect of this work is the connection between mitochondria oxidative stress and neural crest differentiation defects. As presented however, the work lacks mechanism and need to be strengthened before this manuscript is accepted for publication. Below I propose some experiments to strengthen the conclusions of the manuscript.

1) The authors developed a method to detect abasic sites generated by oxidative stress. While useful and informative, there's no validation of this method. How do we know that this method works as intended?

2) The authors should use an alternative experimental approach to confirm that the oxidative stress response is restricted to the mitochondrial genome.

3) If oxidative stress in the mitochondria is the major cause of the craniofacial phenotype, then mitochondrial disfunction should be evident in the Neil1/2 mutant background upon neuroectoderm induction. Experimental evidence for this is lacking.

4) Most of the mechanistic data derived from the interpretation of gene expression analyses. I found these data supportive, but unconvincing. There is a wealth of commercially available antibodies/reagents to study intrinsic apoptosis. The authors should leverage these reagents to solidify their conclusions.

Lastly, many statements throughout the paper are not accurate and therefore misleading to the reader. Also, the authors are very selective of the literature they chose to reference.

Reviewer #2:

In this manuscript, Dandan Han, Christof Niehrs and colleagues describe a mechanism whereby glucosylases NEIL1 and NEIL2 protect cranial neural crest cells (cNCC) against oxidative DNA (particularly mitochondrial DNA) damage. They used a used a series of morpholino-mediated downregulations in *Xenopus*, and Crispr/Cas9 knockouts in the mouse embryonic stem cells and convincingly show that the disruption of NEIL glucosylases (mainly 1 and 2) induces a Tp53 DNA damage response and results in differentiation defects of cNCC. The amount of data is impressive and the quality is very high.

I have only one major point that needs to be addressed experimentally and few other points that should be clarified in the text.

The validation of the mESCs Ko is incomplete/insufficient. Western blots for many targeted proteins (Neil2 and 3). The off-target effect inherent to CRISPR/Cas9 is a real issue both in the literature and in this study as well (see for instance Figure 6—figure supplement 1B that shows that 2 distinct deletions occur in a small region (+ 1kb) amplified by PCR in the triple Ko – Introduction). Whole genome sequencing is the only reliable way to rule out the possibility of off target mutations. At least uncropped gel images for all the KO cells have to be shown.

Reviewer #3:

In this manuscript Han et al. investigate the role of NEIL family bifunctional DNA glycosylases/lyases in *Xenopus* embryos and differentiating mouse embryonic stem cells (mESCs). They find that in *Xenopus* embryos Neil2 deficiency leads to defects in cranial neural crest cell development and link this to an oxidative stress-induced TP53-dependent mitochondrial DNA damage response that leads to apoptosis. They then use mESCs to confirm similar findings in mammalian cells. They suggest a model whereby NEIL1/NEIL2 function to repair and protect the mitochondrial genome against oxidative damages that accompany the metabolic switch upon neural differentiation. The work in this manuscript is, for the most part, well executed and the findings are potentially of broad interest. A problem that persists throughout the work, however, is the conflating of cranial neural crest formation/development with neural differentiation, which is confusing and detracts from the impact of the work. This is particularly odd given that they one late phenotype that the authors examine is craniofacial cartilage, which is clearly not an example of neural differentiation.

From the *Xenopus* work it appears that the main finding is that NEIL2 morphants and pyocyanin treated embryos show a loss of neural crest markers at neurula stages and that this ultimately results in a loss of neural crest derivatives as evidenced by the cartilage phenotype. The authors link this to an increase in tp53, and show that tp53 over-expression also leads to a decrease in neural crest markers. Oddly, the authors conclude from this that "the cNCC defects in Neil2 morphants reflect a "neural-restricted Tp53-response to oxidative DNA damage". However they look at Sox3 expression in the neural plate and it is relatively normal. If the authors wish to extend their conclusions to neural rather than neural crest development then they need to look at a broader range of neural markers, including at later stages. In the absence of this, they can really only make conclusions about neural crest cells.

The experiments with pyocyanin are problematic as the treatment appears to have been from early cleavage stages so any effects on neural crest could be very indirect – what happens if you treat later, for example at gastrulation?

In Figure 1H the authors preclude an effect on proliferation using a western blot for p-H3 but one would not likely detect local effects by this assay – they should do whole mount staining

Tunnel staining in Figure 1J shows increased apoptosis in both neural and non-neural cells but the authors state that cell death is in neural cells.

It is interesting that p53 expression is elevated at the neural folds relative to other tissues. One might think that would render these cells more rather than less protected against oxidative stress than their neighbors. The authors should comment on this. Also what happens to p53 expression in Neil2 morphants or after pyocyanin treatment?

The authors see a decrease in Slug expression after tp53 injection, but not loss of the lineage tracer β-galactosidase, so the cells are presumably still there. This call into question whether the decrease observed is really cell death dependent. They should block apoptosis with Bcl2 and determine if expression is rescued. Also given the authors continued focus on neural differentiation, does p53 injection alter sox3 expression or other neural markers such as neural tubulin?

In the second half of the paper focusing on mESCs the authors interpret their findings as requirement for NEIL function in cNCC development is evolutionarily conserved between amphibians and mammals. However they draw this conclusion without convincing data for neural phenotypes in *Xenopus* and using only one marker each for neural and neural crest (Pax6/Pax3). Multiple markers for each cell type should be examined.

---

## [Author Response]

Essential revisions:1) The authors must drop the claim that there is an effect on neural differentiation in Xenopus or do the work needed to validate this claim. e.g. look at multiple markers at both neural plate stages and later tailbud stages at neural differentiation markers.

We agree and we now emphasize that the phenotype of Neil2-morphant *Xenopus* is specific to cNCC- but not to neural differentiation.

2) The authors must use multiple markers in mESCs to validate effects on neural crest and neural differentiation.

We have now added three additional markers for neural crest (*Hoxa2, Tfapb2, Neurog1*) and neural (*Nestin, Sox1, Pax2*) differentiation supporting our conclusion about cNCC and neural differentiation defects in *Neil1*- and *Neil2*-deficient mESCs. The results are presented in Figure 6—figure supplement 2 and Figure 7—figure supplement 2.

3) The authors should examine tp53 expression in NEIL2 morphants and pyocyanin treated embryos.

We now performed *tp53* expression analysis in stage 14 Neil2 morphants and pyocyanin-treated embryos. Tp53 expression is significantly elevated in both cases albeit less strong when compared to Tp53 target genes (Figure 1F and 3B).

And for pyocyanin treated embryos drug treatment should be done more proximal to the time of neural crest/neural plate formation.

We treated embryos with pyocyanin from stage 8.5 and 10 onwards and with different doses (25 and 80 µM) but could not detect a Tp53 DDR at stage 14/15 nor any strong phenotype at stage 32 (data not shown) with the exception of high dose pyocyanin from stage 8.5 onwards. The time may be too short for sufficient oxidative DNA damage accumulation and full Tp53 cascade induction. In addition, we showed that *neil2* expression is temporally restricted in *Xenopus* embryos until stage 10 (PMID: 26751644) indicative of a critical requirement for Neil2 in lesion repair in early stages prior to neural crest development. Hence, we conclude that to mimic Neil2-deficiency, Pyocyanin treatment needs to start from early stages onwards.

4) More rigorous validation of the CRISPR-Cas9 mutations is needed.

1) We previously validated the knockouts in all generated cell lines by several means:

i) Genotyping PCR on gDNA with primers spanning the anticipated deletion region as presented in Figure 6—figure supplement 1, Figure 7—figure supplement 1, Figure 8—figure supplement 1, Figure 9—figure supplement 1 and Figure 9—figure supplement 2 (now also shown as uncropped agarose gels).

ii) Western blot analysis for endogenous proteins if antibodies were available and working (NEIL1 in Figure 6—figure supplement 1 and Figure 7—figure supplement 1, APEX1 in Figure 8—figure supplement 1, and TDG in Figure 9—figure supplement 1 and Figure 9—figure supplement 2).

iii) RT-qPCR for expression analysis of the targeted exons (Neil2 and Neil3, Figure 6—figure supplement 1, Figure 7—figure supplement 3). Together, these analyses unambiguously confirmed the anticipated knockouts in the respective lines.

2) We now analyzed with CRISPOR (PMID: 27380939) off-target probability and potential off-target regions of the 6 gRNAs used to generate the *Neil1, Neil2* and *Apex1* knockout lines. These are the cell lines that showed neural and cNCC differentiation defects and were subjected to RNAseq analysis as undifferentiated mESCs. All 6 gRNAs were scored >75 in the CFD specificity score (scale from 0 – 100, the higher the specificity score, the lower the probability for off-targets in the genome). In addition, we found among the potential off-target regions 25 genes with exonic matches for the two *Neil1* gRNAs, 14 for *Neil2* and 12 for *Apex1* all of which had between 3-4 base mismatches. Frameshift mutations are frequently accompanied by RNA level deregulation due to non-sense mediated decay or compensatory upregulation, respectively. However, only 4 of the 51 total candidates showed slight (log2FC <1.0) RNA deregulation in the respective knockout lines compared to control mESCs (data not shown).

3) Most importantly, stable expression of *NEIL1* and *NEIL2* in the respective *Neil1* and *Neil2* knockout lines rescued the neural and neural crest differentiation defects (Figure 7—figure supplement 3) showing the specificity of the knockout approach.

5) The authors need to provide additional evidence demonstrating that the oxidative stress response directly causes the phenotype and originates in the mitochondria. To me this is the most novel aspect of the work.

1) We now performed L-ascorbic acid (Vitamin C) antioxidant treatment of *neil2* MO-injected *Xenopus* embryos. Vitamin C partially rescued the severe phenotype induced by Neil2-deficiency supporting the link between the neural crest phenotype and oxidative stress (Figure 3G).

2) We now confirmed elevated mtDNA damage in *Neil1* and *Neil2*-deficient EBs + RA (Figure 11C) by a mass spectrometry-independent approach based on quantitative PCR for damage detection (PMID: 28286206). In this long-range PCR, levels of DNA damage are monitored based on the fact that DNA lesions inhibit DNA polymerase and slow down accumulation of the PCR product. Therefore, the rate of product amplification is inversely proportional to the number of damaged DNA molecules. The results shows damage accumulation notably under RA-induced neural differentiation but not in undifferentiated mESCs, corroborating our conclusions.

3) Mutations on mtDNA and subsequent mitochondrial dysfunction result in compensatory upregulation of mitochondrial OXPHOS genes (PMID: 10438462, 19389473). Hence, we monitored the expression of a number of mitochondrial OXPHOS genes in both model systems, *Xenopus* and mESCs, and found a systematic upregulation of these genes in Neil2-deficient *Xenopus* embryos as well as *Neil1-* and *Neil2*-deficient mESCs during neural differentiation (Figure 2C and 11D). The result supports the notion that the stress response leads to mitochondrial dysfunction following mtDNA damage.

4) We have validated the mass spectrometry-based quantification of abasic sites (PMID: 28715893) and oxidative base damages using synthetic oligonucleotides with defined amounts of lesions. The result confirms the approach to function as intended (presented in Figure 11—figure supplement 1C). Hence, the quantification of base damages on cellular DNA (Figure 11B) is reliable.

The full reviews are attached to provide further details, which we advise you to address if possible.Reviewer #1:[…] 1) The authors developed a method to detect abasic sites generated by oxidative stress. While useful and informative, there's no validation of this method. How do we know that this method work as intended?

We have validated the mass spectrometry-based quantification of abasic sites (PMID: 28715893) and oxidative base damages using synthetic oligonucleotides with defined amounts of lesions. The result confirms the approach to function as intended (presented in Figure 11—figure supplement 1C). Hence, the quantification of base damages on cellular DNA (Figure 11B) is reliable.

2) The authors should use an alternative experimental approach to confirm that the oxidative stress response is restricted to the mitochondrial genome.

We added further evidence for elevated mtDNA damage in *Neil1* and *Neil2*-deficient EBs + RA (Figure 11C) by a mass spectrometry-independent approach based on quantitative PCR for damage detection (PMID: 28286206). In this long-range PCR, levels of DNA damage are monitored based on the fact that DNA lesions inhibit DNA polymerase and slow down accumulation of the PCR product. Therefore, the rate of product amplification is inversely proportional to the number of damaged DNA molecules. The results shows damage accumulation notably under RA-induced neural differentiation but not in undifferentiated mESCs, corroborating our conclusions.

3) If oxidative stress in the mitochondria is the major cause of the craniofacial phenotype, then mitochondrial disfunction should be evident in the Neil1/2 mutant background upon neuroectoderm induction. Experimental evidence for this is lacking.

We now show elevated mt gene expression indicative of mitochondrial dysfunction in both model systems, *Xenopus* and mESCs (Figure 2C and 11D).

4) Most of the mechanistic data derived from the interpretation of gene expression analyses. I found these data supportive, but unconvincing. There is a wealth of commercially available antibodies/reagents to study intrinsic apoptosis. The authors should leverage these reagents to solidify their conclusions.We did utilize a variety of antibodies (targeting Caspase-3, Caspase-7, Caspase-9, AIF, Cytochrome C) to confirm an intrinsic apoptosis pathway in EB + RA differentiated cells. However, by Western blot we could not detect substantial activation or translocation of these apoptotic factors in *Neil*-knockout- as compared to control clones (Figure 12—figure supplement 2B). This could be due to insufficient sensitivity in a whole-population analysis with a fraction of cells being affected. Instead, we detected and confirmed a requirement for Caspase-3 (cleaved and uncleaved) during neuronal differentiation (Figure 12 —figure supplement 2A) indicative of a systemic regulation of apoptotic factors. Moreover, TP53 target gene activation in EB + RA cells was moderate (< 2-fold) though significant as assessed by RT-qPCR analysis (Figure 12D and Figure 12—figure supplement 1C). Importantly, we now show a rescue of the Neil2 MO-induced phenotype by *bcl2l1* overexpression in *Xenopus* embryos confirming ongoing intrinsic apoptosis by *Neil2*-deficiency (Figure 2A-B).Lastly, many statements throughout the paper are not accurate and therefore misleading to the reader. Also, the authors are very selective of the literature they chose to reference.

We apologize for the missing accuracy and tried our best to improve the manuscript text and references, see below.

Reviewer #2:[…] I have only one major point that needs to be addressed experimentally and few other points that should be clarified in the text.The validation of the mESCs Ko is incomplete/insufficient. Western blots for many targeted proteins (Neil2 and 3). The off-target effect inherent to CRISPR/Cas9 is a real issue both in the literature and in this study as well (see for instance Figure 6—figure supplement 1B that shows that 2 distinct deletions occur in a small region (+ 1kb) amplified by PCR in the triple Ko – Introduction). Whole genome sequencing is the only reliable way to rule out the possibility of off target mutations. At least uncropped gel images for all the KO cells have to be shown (I'm happy to hear from other reviewers and editors on this point).

We agree that potential off-target effects are an issue in CRISPR/Cas9 genome editing, and did further analysis.

1) We previously validated the knockouts in all generated cell lines by several means:

i) Genotyping PCR on gDNA with primers spanning the anticipated deletion region as presented in Figure 6—figure supplement 1, Figure 7—figure supplement 1, Figure 8—figure supplement 1, Figure 9—figure supplement 1 and Figure 9—figure supplement 2 (now also shown as uncropped agarose gels).

ii) Western blot analysis for endogenous proteins if antibodies were available and working (NEIL1 in Figure 6—figure supplement 1 and Figure 7—figure supplement 1, APEX1 in Figure 8—figure supplement 1, and TDG in Figure 9—figure supplement 1 and Figure 9—figure supplement 2).

iii) RT-qPCR for expression analysis of the targeted exons (Neil2 and Neil3, Figure 6—figure supplement 1, Figure 7—figure supplement 3). Together, these analyses unambiguously confirmed the anticipated knockouts in the respective lines.

2) We now analyzed with CRISPOR (PMID: 27380939) off-target probability and potential off-target regions of the 6 gRNAs used to generate the *Neil1, Neil2* and *Apex1* knockout lines. These are the cell lines that showed neural and cNCC differentiation defects and were subjected to RNAseq analysis as undifferentiated mESCs. All 6 gRNAs were scored >75 in the CFD specificity score (scale from 0 – 100, the higher the specificity score, the lower the probability for off-targets in the genome). In addition, we found among the potential off-target regions 25 genes with exonic matches for the two *Neil1* gRNAs, 14 for *Neil2* and 12 for *Apex1* all of which had between 3-4 base mismatches. Frameshift mutations are frequently accompanied by RNA level deregulation due to non-sense mediated decay or compensatory upregulation, respectively. However, only 4 of the 51 total candidates showed slight (log2FC <1.0) RNA deregulation in the respective knockout lines compared to control mESCs (data not shown).

3) It is not surprising to see two distinct bands when deleting a target region of a gene in both alleles with two gRNAs since the error prone non-homologous end-joining will most likely produce different repair outcomes at the two allelic loci. Two PCR bands are also detectable with the *Neil1* single-knockout clone #11 (Figure 7—figure supplement 1A). Such a result is not a sign of off-target effects of the gRNAs. Notably, most of the genotyping PCRs generated apparently a unique deletion band; however, it is more likely that such bands consist of two different PCR products with similar length not resolvable by an agarose gel. To this end, we sequenced the apparent single band PCR products of all three *Neil2* single knockout clones (as shown in Figure 7—figure supplement 1A) and could indeed reveal twice two distinct sequences:

Neil2 #1

5’ACACTATAGTGATTAGTATTTGCTG---615bp---ATGACCCCACGCAGGATCAGAAGAA 3‘

5’TAGTATTTGCTGTGTGGTTAAGGCT---587bp---AAAAGGATGGACCTGATGACCCCAC 3‘

Neil2 #11

5’GTATTTGCTGTGTGGTTAAGGCTGA---586bp---AAAGGATGGACCTGATGACCCCACG 3‘

Neil2 #14

5’ AAGACTCTGTCTTAACCAACCAAAC---695bp---AAGGATGGACCTGATGACCCCACGC 3‘

5’ TGATTAGTATTTGCTGTGTGGTTAA---594bp---AGGATGGACCTGATGACCCCACGCA 3‘

4) Most importantly, stable expression of *NEIL1* and *NEIL2* in the respective *Neil1* and *Neil2* knockout lines rescued the neural and neural crest differentiation defects (Figure 7—figure supplement 3) showing the specificity of the knockout approach.

Reviewer #3:[…] A problem that persists throughout the work, however, is the conflating of cranial neural crest formation/development with neural differentiation, which is confusing and detracts from the impact of the work. This is particularly odd given that they one late phenotype that the authors examine is craniofacial cartilage, which is clearly not an example of neural differentiation.From the Xenopus work it appears that the main finding is that NEIL2 morphants and pyocyanin treated embryos show a loss of neural crest markers at neurula stages and that this ultimately results in a loss of neural crest derivatives as evidenced by the cartilage phenotype. The authors link this to an increase in tp53, and show that tp53 over-expression also leads to a decrease in neural crest markers. Oddly, the authors conclude from this that "the cNCC defects in Neil2 morphants reflect a "neural-restricted Tp53-response to oxidative DNA damage". However they look at Sox3 expression in the neural plate and it is relatively normal. If the authors wish to extend their conclusions to neural rather than neural crest development then they need to look at a broader range of neural markers, including at later stages. In the absence of this, they can really only make conclusions about neural crest cells.

We agree and we now emphasize that the phenotype of Neil2-morphant *Xenopus* is specific to cNCC- but not to neural differentiation.

The experiments with pyocyanin are problematic as the treatment appears to have been from early cleavage stages so any effects on neural crest could be very indirect – what happens if you treat later, for example at gastrulation?We treated embryos with pyocyanin from stage 8.5 and 10 onwards and with different doses (25 and 80 µM) but could not detect a Tp53 DDR at stage 14/15 nor any strong phenotype at stage 32 (data not shown) with the exception of high dose pyocyanin from stage 8.5 onwards. The time may be too short for sufficient oxidative DNA damage accumulation and full Tp53 cascade induction. In addition, we showed that *neil2* expression is temporally restricted in *Xenopus* embryos until stage 10 (PMID: 26751644) indicative of a critical requirement for Neil2 in lesion repair in early stages prior to neural crest development. Hence, we conclude that to mimic Neil2-deficiency, Pyocyanin treatment needs to start from early stages onwards.In Figure 1H the authors preclude an effect on proliferation using a western blot for p-H3 but one would not likely detect local effects by this assay – they should do whole mount staining.

We performed p-H3 whole mount immunoassays in Neil2 morphant embryos in our previous study (PMID: 26751644). The result is congruent with the western blot shown in the current study.

Tunnel staining in Figure 1J shows increased apoptosis in both neural and non-neural cells but the authors state that cell death is in neural cells.

We did not conclude from Figure 1J that apoptosis is restricted to neural cells: “Whole mount TUNEL assay of unilaterally neil2 MO-injected embryos confirmed elevated apoptosis by Neil2-deficiency in stage 16 embryos (Figure 1J).” However, we concluded from the western blot shown in Figure 1I that cleaved Caspase-3 levels (apoptosis) were elevated in neural- compared to non-neural tissues.

It is interesting that p53 expression is elevated at the neural folds relative to other tissues. One might think that would render these cells more rather than less protected against oxidative stress than their neighbors. The authors should comment on this.

We state in the Discussion:

“Neurogenesis is accompanied by a metabolic switch from glycolysis to oxidative phosphorylation, a concomitant increase in mass and number of mitochondria per cell, and hence escalated oxidative stress (Khacho and Slack, 2018). […] Hence, cells with high basal Tp53 expression will reach detrimental Tp53 levels more easily upon stress.”

Also what happens to p53 expression in Neil2 morphants or after pyocyanin treatment?

Tp53 expression is significantly elevated in both cases albeit less strong when compared to Tp53 target genes (Figure 1F and 3B).

The authors see a decrease in Slug expression after tp53 injection, but not loss of the lineage tracer β-galactosidase, so the cells are presumably still there. This call into question whether the decrease observed is really cell death dependent. They should block apoptosis with Bcl2 and determine if expression is rescued. Also given the authors continued focus on neural differentiation, does p53 injection alter sox3 expression or other neural markers such as neural tubulin?

*Bcl2l1* (the *Xenopus Bcl2* homologue) mRNA injection indeed rescued *Neil2* MO-triggered malformations as well as elevated levels of cleaved Caspase-3 in neural tissues (new Figure 2A-B) supporting induced apoptosis by Neil2-deficiency in *Xenopus*, and a functional link between apoptosis and the observed malformations.

In the second half of the paper focusing on mESCs the authors interpret their findings as requirement for NEIL function in cNCC development is evolutionarily conserved between amphibians and mammals. However they draw this conclusion without convincing data for neural phenotypes in Xenopus and using only one marker each for neural and neural crest (Pax6/Pax3). Multiple markers for each cell type should be examined.

We have now added three additional markers for neural crest (*Hoxa2, Tfapb2, Neurog1*) and neural (*Nestin, Sox1, Pax2*) differentiation supporting our conclusion about cNCC and neural differentiation defects in *Neil1*- and *Neil2*-deficient mESCs. The results are presented in Figure 6—figure supplement 2 and Figure 7—figure supplement 2.